# Association study in African-admixed populations across the Americas recapitulates asthma risk loci in non-African populations

Michelle Daya et al.[#]

Asthma is a complex disease with striking disparities across racial and ethnic groups. Despite its relatively high burden, representation of individuals of African ancestry in asthma genome-wide association studies (GWAS) has been inadequate, and true associations in these underrepresented minority groups have been inconclusive. We report the results of a genome-wide meta-analysis from the Consortium on Asthma among African Ancestry Populations (CAAPA; 7009 asthma cases, 7645 controls). We find strong evidence for association at four previously reported asthma loci whose discovery was driven largely by non-African populations, including the chromosome 17q12–q21 locus and the chr12q13 region, a novel (and not previously replicated) asthma locus recently identified by the Trans-National Asthma Genetic Consortium (TAGC). An additional seven loci reported by TAGC show marginal evidence for association in CAAPA. We also identify two novel loci (8p23 and 8q24) that may be specific to asthma risk in African ancestry populations.

---

Asthma is a complex disease where the interplay between genetic factors and environmental exposures controls susceptibility and disease progression. In the U.S., ethnic minorities are disproportionally affected by asthma. For example, African Americans and Puerto Ricans have higher asthma-related morbidity and mortality rates compared to European Americans[1–3]. In addition to environmental, cultural, and socioeconomic risk factors, genetic factors, possibly from a common background ancestry, likely underlie some of these disparities in the health burden of asthma in the U.S. Despite the relatively high burden of disease, representation of African ancestry populations in asthma genome-wide association studies (GWAS) has been limited, and in GWAS's performed to date including individuals of African ancestry, the samples have been modest and underpowered to detect true associations. For example, the largest asthma GWAS focused solely on African ancestry populations included only 819 asthma cases[4]. Recently, the Trans-National Asthma Genetic Consortium (TAGC) reported 18 asthma-associated loci based on a meta-analysis of 142,486 subjects, but only 2149 cases and 6055 controls included in this study were of African ancestry. Only two genome-wide significant associations[5,6] have been reported for asthma from GWAS conducted in African ancestry populations to date[4–8]. The discovery of genetic risk factors for asthma in African ancestry populations has been further hampered by lack of representation of African ancestry in imputation reference panels, and legacy commercial genotyping arrays that have until very recently not provided adequate coverage of linkage disequilibrium (LD) patterns in African ancestry populations.

To address these disparities in asthma genetics research and the paucity of information on African genetic diversity, we previously established the Consortium on Asthma among African ancestry Populations in the Americas (CAAPA)[9]. Because of the lack of adequate representation of African ancestry in imputation reference panels, we first performed whole-genome sequencing (WGS) of samples collected from 880 individuals who self-reported African ancestry from 19 North, Central and South American and Caribbean populations (446 individuals from nine African American populations, 43 individuals from Central America, 121 individuals from three South American populations, and 197 individuals from four Caribbean populations), as well as individuals from continental West Africa (45 Yoruba-speaking individuals from Ibadan, Nigeria and 28 individuals from Gabon). These whole-genome sequences were made publicly available through dbGAP (accession code phs001123.v1.p1) and were incorporated into the reference panel on the Michigan imputation server (a free genotype imputation service, https://imputationserver.sph.umich.edu).

Previously we performed coverage analysis of the novel variation identified in the CAAPA sequence data, and found only 69% of common SNP variants and 41% of low-frequency SNP variants identified by CAAPA can be tagged by traditional GWAS arrays at $r^2 \geq 0.8$[10]. In addition, lack of coverage of low frequency variants (minor allele frequency [MAF] between 0.01–0.05) in GWAS arrays negatively impacts the imputation of low frequency variants. To address these issues, we used the CAAPA sequence data to develop the African Diaspora Power Chip (ADPC) in partnership with Illumina, Inc., a gene-centric SNP genotyping array designed to complement commercially available genome-wide chips, thereby improving tagging and coverage of African ancestry genetic variation[10]. The array included ~495,000 SNPs, with a MAF enriched for low frequency variants. Subsequently, the content of the ADPC was incorporated into Illumina's Multi-Ethnic Global Array (MEGA)[10].

Using the ADPC, we genotyped CAAPA participants from nine studies (seven African American studies, one African

Caribbean [Barbados], and one South American [Puerto Rico]). We combined the ADPC data with existing commercial genome-wide genotype data, and imputed additional genotypes using the CAAPA reference panel. Subsequently we used the MEGA to genotype additional African ancestry asthma studies with no previously existing genome-wide genotype data (one African American, three South American, and two Caribbean studies), and similarly imputed genotypes on these subjects using the CAAPA reference panel. Sample populations, their ascertainment, and clinical characteristics are described in detail in the Supplementary Note 1, Table 1 and Supplementary Table 1. We then used these data to perform a GWAS of asthma in individuals of African ancestry (7009 asthmatic cases and 7645 controls). We also performed admixture mapping, a technique that leverages local ancestry to identify regions of the genome where ancestry from a particular ancestral population is inherited more frequently among affected compared to unaffected individuals.

Our GWAS results recapitulate 11 of the 18 loci recently reported by TAGC, including the chromosome 9p24 (IL33), 15q22 (RORA), and 17q12–q21 loci, as well as a locus on chromosome 12q13 (STAT6) reported as novel but not replicated by TAGC. We identify two loci on chromosome 8 not previously reported as being associated with asthma, and one local ancestry peak on chromosome 6q22 is genome-wide significant in our admixture mapping study. We speculate that the common asthma-associated variants on chromosome 8p23 implicate the ARHGEF10 or MYOM2 genes, while a low frequency protective variant on chromosome 8q24 is intronic to TATDN1, a gene with increased expression in human airway smooth muscle cells stimulated with interleukin. The TCF21 gene, which has previously been shown to be differentially expressed in bronchial biopsies of asthmatics compared to controls, is a likely candidate in the admixture mapping peak. However, further replication efforts are necessary to provide robust evidence of replication for these chromosome 6 and 8 loci.

## Results

**Association analysis.** Studies included in the asthma association analysis represent a diverse spectrum of African ancestry (Fig. 1a, Supplementary Fig. 1, Supplementary Table 2) with median African ancestry proportions in non-asthmatics as low as 0.17 in subjects from Puerto Rico (GALA II) and as high as 0.90 in subjects from Jamaica and Barbados (JAAS and BAGS). In addition, the studies had different objectives and differed in potentially relevant factors such as age of onset and diagnostic criteria (see Supplementary Note 1 and Supplementary Table 1). For this reason, we performed tests for association separately for each dataset, and then combined the results using MR-MEGA, a novel meta-analysis approach that models allelic effects as a function of axes of genetic variation[11]. In this way, heterogeneous associations across genetically distinct populations are not penalized, and the degree of heterogeneity due to ancestry, as well as residual effects due to differences in study design, can be assessed. The results of the genome-wide meta-analysis are summarized in Fig. 1b and c, and associations with MR-MEGA association $p < 10^{-6}$ (the same cut-off employed by the EVE GWAS[6]) are summarized in Table 2 and Supplementary Table 9. There were seven loci with associations smaller than this threshold. Associations at two loci were genome-wide significant ($p < 5 \times 10^{-8}$), including a locus at chromosome 8p23 not previously reported by any asthma GWAS[7,12], and the chromosome 17q12–q21 locus, which is regarded as one of the most consistent asthma association findings to date[13]. Two of the loci with $p < 10^{-6}$ (but not attaining conventional genome-wide significance) were reported recently by TAGC in the multi-ethnic

**Table 1 Studies included in the CAAPA association analysis**

| Population | Location | Study | GWAS platform | Non-asthmatics | Asthmatics | Total | Nr SNPs |
|---|---|---|---|---|---|---|---|
| African American | Baltimore | BASS | Illumina MEGA | 216 | 135 | 351 | 12,403,613 |
| | Baltimore | GRAAD(1) | ADPC + Illumina HumanHap 650Y | 385 | 396 | 781 | 15,486,076 |
| | Baltimore | GRAAD(2) | ADPC + Illumina OMNI 2.5 | 23 | 65 | 88 | 7,496,303 |
| | Chicago | CAG | ADPC + Illumina HumanHap1M | 156 | 114 | 270 | 11,604,736 |
| | Detroit | SAPPHIRE | ADPC[a] + Affymetrix Axiom AFR | 566 | 1325 | 1891 | 18,768,360 |
| | Jackson | JHS(1) | ADPC + Affymetrix 6.0 | 283 | 44 | 327 | 12,124,436 |
| | Jackson | JHS(2) | ADPC + Affymetrix 6.0 | 546 | 101 | 647 | 14,777,976 |
| | San Francisco | SAGE II | ADPC + Affymetrix Axiom LAT | 691 | 1001 | 1692 | 18,008,099 |
| | Washington | HUFS | ADPC + Affymetrix 6.0 | 1527 | 303 | 1830 | 18,102,295 |
| | Winston-Salem | SARP | ADPC + Illumina HumanOmniExpress + HumanHap1M | 45 | 302 | 347 | 12,281,618 |
| | | | | **4438** | **3786** | **8224** | |
| Barbados | Barbados | BAGS | ADPC + Illumina HumanHap 650Y | 338 | 282 | 620 | 14,546,148 |
| Brazil | Condé | BIAS | Illumina MEGA | 426 | 194 | 620 | 13,663,002 |
| Brazil | Salvador | ProAR | Illumina MEGA | 346 | 761 | 1107 | 15,963,442 |
| Colombia | Cartagena | PGCA | Illumina MEGA | 488 | 664 | 1152 | 15,093,107 |
| Honduras | Honduras | HONDAS | Illumina MEGA | 249 | 254 | 503 | 12,895,713 |
| Jamaica | Jamaica | JAAS | Illumina MEGA | 507 | 167 | 674 | 14,792,220 |
| Puerto Rico | Puerto Rico | GALA II | ADPC + Affymetrix Axiom LAT | 853 | 901 | 1754 | 15,125,996 |
| | | | | **7645** | **7009** | **14,654** | **23,328,733** |

The table summarizes the geographical location and final number of subjects and imputed SNPs included in the asthma association analysis. Bold numbers are the sum of the numbers above.
[a]ADPC data available for 730 asthmatics only

meta-analysis of 23,948 asthmatics and 118,538 controls[6]. The remaining three loci involved low-frequency SNPs (MAF between 0.01–0.05), and the accuracy of SNP imputation could only be verified for one locus on chromosome 8q24 (Supplementary Table 10).

**Associations novel to CAAPA.** Several SNPs intronic to a gene encoding a long non-coding RNA on chromosome 8p23 have MR-MEGA association $p < 10^{-6}$, and two of these associations were genome-wide significant. While this association was observed in multiple African American samples, the strongest effects were observed in three non-U.S. studies from Barbados (BAGS), Cartagena, Colombia (PGCA), and Puerto Rico (GALA II; Supplementary Fig. 15). While the associated SNPs in this region do not overlap with any expression quantitative trait loci (eQTLs) in the publicly available databases we mined, long-range chromatin interaction and expression data in relevant tissues (lymphoblastoid cells, fetal lung fibroblast cells, and lung) implicate two different genes, *ARHGEF10* and *MYOM2*, ~600–800 KB downstream from the most significant SNP rs13277810 that potentially explain these observed associations (Supplementary Note 13, Supplementary Fig. 19)[14–16]. Expression of *ARHGEF10* has been associated with exacerbations in chronic obstructive pulmonary disease[17], and this gene may also be a target of the well-known type-2 inflammation cytokine interleukin 33[18], while genetic variants in *MYOM2* are predictive of lung function in an isolated European ancestry (Hutterite) population[19]. We tested whether SNP rs13277810 is associated with asthma using African American genetic data from three cardiovascular studies as well as African American and Hispanic genetic data from the BioMe biobank. Although there was strong evidence for replication of this SNP association in BioMe Hispanics (SNP association Z-score $p = 7.15 \times 10^{-4}$, with consistent effect direction), the SNP was not associated with asthma in any of the other studies, and the TAGC asthma GWAS summary statistics also did not support this association in Europeans (Table 3). Most CAAPA asthmatics had childhood onset asthma (Supplementary Table 1), which is likely not the case for the cardiovascular disease studies. Roughly a third of BioMe Hispanics are Puerto Rican (Supplementary Note 2), and Puerto Rican asthmatics are likely to have childhood onset asthma[20]. We therefore re-tested this association in CAAPA, and compared results when including versus excluding adult onset asthmatics. Because the effect size of the association decreased when adult onset asthmatics were excluded (Supplementary Table 17), we could not confirm that age of asthma onset accounts for the lack of replication. Despite the potential biological plausibility of this chromosome 8p23 association, it may be a false positive. Alternatively, lack of replication could be due to differences in study design (ascertainment of subjects and asthma case-control classification), stronger effects in non-U.S. African ancestry populations, and the small number of asthmatics available in the replication data.

The remaining three loci yielding MR-MEGA association $p < 10^{-6}$ have not previously been reported by any asthma GWAS, and the identified SNPs were all low frequency variants (0.01 < MAF < 0.05). According to allele frequencies from the 1000 Genomes Project (TGP), these SNPs are present in African but not European populations. The imputation quality of the most significant SNPs at these regions is only moderate (median RSQ statistics of 0.79, 0.61, and 0.77 for rs114647118 [8q24], rs73952947 [18q12], and rs73595000 [19q13], respectively, see Supplementary Data 1); therefore we performed TaqMan genotyping to confirm results in the JAAS (Jamaica) and BAGS (Barbados) samples (Supplementary Table 10). The veracity of imputation and asthma association could only be verified for rs114647118 on chromosome 8q24 (Supplementary Tables 10 and 11). Carriers of the minor allele at SNP rs114647118 were less likely to have asthma, and this pattern was observed in all CAAPA studies (Supplementary Fig. 15). SNP rs114647118 is intronic to *TATDN1*, a gene with increased expression in human airway smooth muscle cells stimulated with interleukin 17A[21]. However, this association did not replicate in the three African American cardiovascular studies (Table 3). Unfortunately, data were not available in TAGC or BioMe to test for replication.

**Comparison to previous asthma GWAS.** We compared results from the recent TAGC multi-ethnic meta-analysis, the largest and most definitive GWAS performed in observed asthma cases and controls to date, with the CAAPA meta-analysis[7]. TAGC reported 18 loci associated with asthma, categorizing nine loci as known asthma susceptibility genes (known), five as new asthma loci (new), two as new signals at loci previously associated with asthma in ancestry-specific populations (ancestry-specific), plus

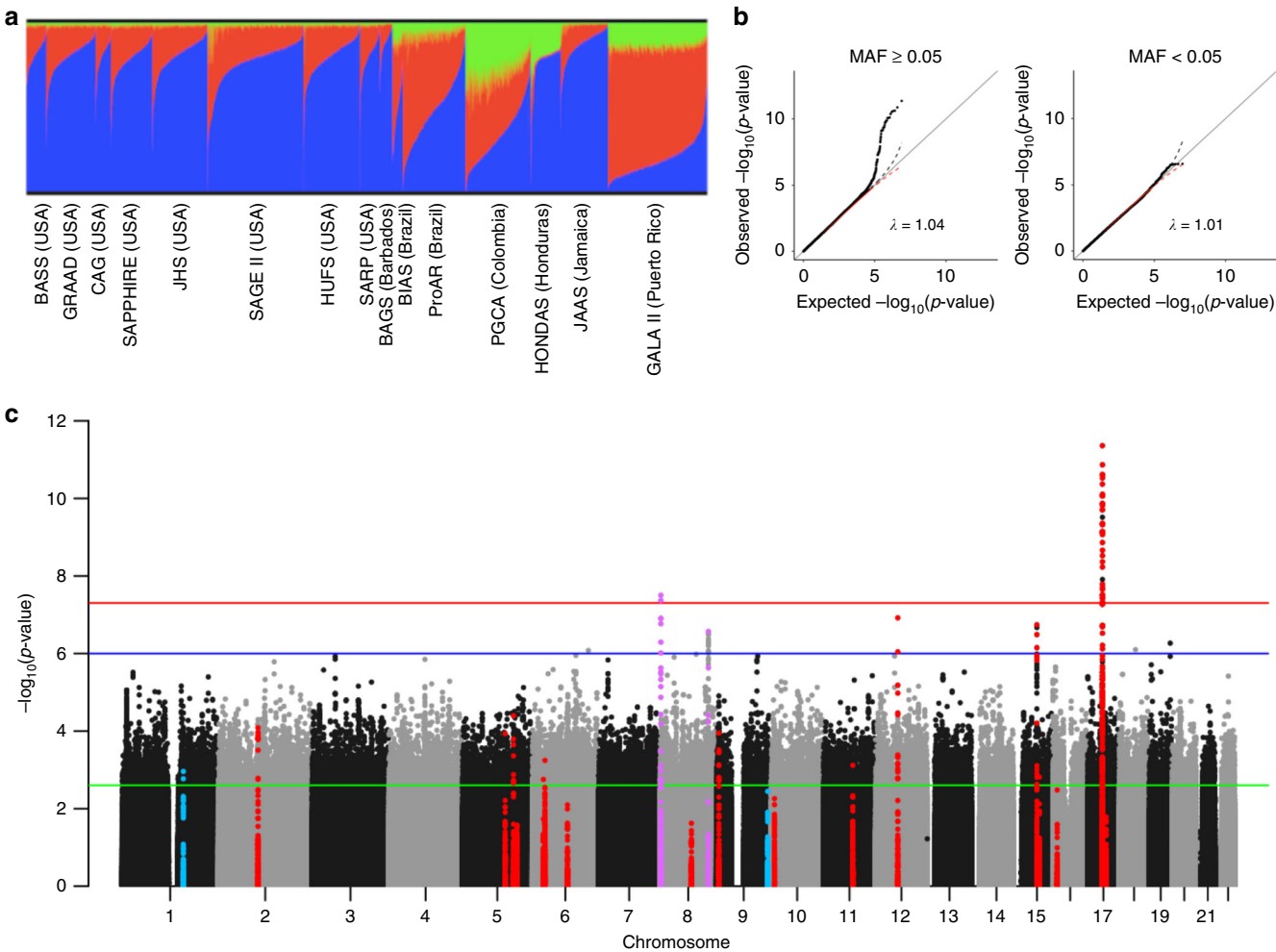

**Fig. 1** Summary of CAAPA ancestry and meta-analysis results. **a** CAAPA ADMIXTURE estimates: this panel summarizes the genome-wide proportions of ancestry for K = 3 populations, as estimated by the software program ADMIXTURE. A combined dataset of 20,482 overlapping and linkage disequilibrium pruned SNPs in 84 African (YRI, blue) and 84 European (CEU, red) 1000 Genomes Project phase 3 subjects, 43 Native American (green) subjects and 12,223 putatively unrelated CAAPA subjects were used to estimate these ancestry proportions. **b** QQ plot of the meta-analysis p-values: the plots in this panel are stratified by minor allele frequency (MAF) for low frequency and common SNPs. Inflation factors were calculated by transforming MR-MEGA association p-values to 1 degree of freedom (df) Chi-square statistics, and dividing the median of these statistics by the median of the theoretical Chi-square (1 df) distribution. The dashed black and red lines represent the upper and lower 95% confidence interval. **c** Manhattan plot of the meta-analysis p-values: the red, blue, and green horizontal lines in the Manhattan plot represent significant (MR-MEGA association $p < 5 \times 10^{-8}$), suggestive (MR-MEGA association $p < 10^{-6}$), and candidate gene (MR-MEGA association $p < 2.6 \times 10^{-3}$) value thresholds, respectively. The candidate gene threshold is a Bonferroni-adjusted alpha level for 20 tests (1 locus from EVE, 1 locus from eMERGE, and 18 loci from TAGC). Windows of ±10 KB around the lead SNP at each selected locus are colored blue (EVE and eMERGE loci), red (TAGC loci), and purple (CAAPA loci with lead SNPs having $p < 10^{-6}$). A larger window of ±200 KB is shown for the chromosome 17q12–21 locus

two as asthma signals previously reported for asthma with hay fever (asthma + hay fever) (Table 2 and Supplementary Table 12)[7]. Because some of the CAAPA studies were included in the TAGC asthma meta-analysis (of the 2149 asthmatics and 6055 non-asthmatics in TAGC, 1601 asthmatics and 2375 non-asthmatics were from studies included in the CAAPA discovery, and 548 asthmatics and 3680 non-asthmatics were from studies included in the CAAPA replication), and to contrast African and European asthma susceptibility loci, we compared CAAPA associations with those observed in TAGC Europeans. Only two of the TAGC loci, on chromosome 12q13 and 5q31.1, were not genome-wide significant in TAGC Europeans as judged by their random effect p-values ($p = 1.6 \times 10^{-7}$ and $p = 1.6 \times 10^{-6}$, respectively), although the fixed-effect p-values for these associations were genome-wide significant ($p = 5.5 \times 10^{-9}$ and $p = 8.5 \times 10^{-10}$, respectively). Both

these loci were not reported by any prior GWAS of asthma, and their p-values in the other ethnic groups represented in TAGC were marginal ($p = 0.05$, 0.22, and 0.60 for the chromosome 12q13 TAGC lead SNP and $p = 0.04$, $5.5 \times 10^{-3}$, and 0.27 for the chromosome 5q31.1 TAGC lead SNP in Africans, Japanese, and Latinos, respectively). We therefore conclude the genome-wide significance of these loci in the TAGC multi-ethnic meta-analysis is largely due to their strong associations observed in subjects of European ancestry.

TAGC summary statistics were merged with the CAAPA meta-analysis results, and associations of 810 SNPs with significant fixed-effect p-values in TAGC Europeans, Bonferroni corrected for the number of overlapping SNPs in the merged dataset, were assessed for replication in the CAAPA meta-analysis. Three known asthma loci replicated in CAAPA, after a Bonferroni

**Table 2 Lead SNP summary**

| GWAS | Locus | Descr. | rsID | hg19 position | Genes | RA/EA | CAAPA | | | TAGC | | |
|---|---|---|---|---|---|---|---|---|---|---|---|---|
| | | | | | | | EAF | OR [95% CI] | P | EAF | OR [95% CI] | P |
| CAAPA | 8p23 | new | rs13277810 | 2,550,802 | LOC101927815 (intronic) | C/T | 0.12 | 1.26 [1.17-1.37] | $3.2\times10^{-8}$ | 0.19 | | |
| | 8q24 | new | rs114647118 | 125,531,353 | TATDN1 (intronic) | C/T | 0.01 | 0.48 [0.37-0.63] | $2.7\times10^{-7}$ | 0.00 | | |
| | 12q13 | known | rs3122929 | 57,509,102 | STAT6, LRP1 (intergenic) | C/T | 0.27 | 1.17 [1.10-1.24] | $9.1\times10^{-7}$ | 0.34 | | |
| | 15q22 | known | rs10519067 | 61,068,347 | RORA (intronic) | G/A | 0.29 | 0.85 [0.80-0.90] | $1.8\times10^{-7}$ | 0.14 | 0.89 [0.85-0.92] | $8.3\times10^{-9}$ |
| | 17q12-21 | known | rs907092 | 37,922,259 | ORMDL3,GSDMB, ZPBP2,ERBB2 | G/A | 0.20 | 0.80 [0.75-0.85] | $4.3\times10^{-12}$ | 0.47 | 1.08 [1.05-1.11] | $3.3\times10^{-9}$ |
| TAGC | 2q12 | known | rs1420101 | 102,957,716 | IL1RL1,IL1RL2, IL18R1 | C/T | 0.33 | 1.07 [1.02-1.13] | 0.03 | 0.37 | 1.12 [1.10-1.15] | $9.1\times10^{-20}$ |
| | 5q22.1 | known | rs10455025 | 110,404,999 | SLC25A46,TSLP | A/C | 0.12 | 1.11 [1.01-1.22] | 0.05 | 0.34 | 1.15 [1.12-1.18] | $2.0\times10^{-25}$ |
| | 5q31 | known | rs20541 | 131,995,964 | IL13,RAD50,IL4 | A/G | 0.79 | 0.95 [0.89-1.02] | 0.27 | 0.79 | 0.89 [0.86-0.91] | $1.4\times10^{-14}$ |
| | 6p22 | new | rs1233578 | 28,712,247 | GPX5,TRIM27 | A/G | 0.35 | 1.02 [0.97-1.08] | 0.32 | 0.13 | 1.11 [1.07-1.15] | $5.3\times10^{-9}$ |
| | 6p21 | known | rs9272346 | 32,604,372 | HLA-DRB1, HLA-DQA1 | G/A | 0.56 | 1.07 [1.02-1.13] | 0.03 | 0.56 | 1.16 [1.13-1.19] | $4.8\times10^{-28}$ |
| | **9p24** | known | rs992969 | 6,209,697 | RANBP6,IL33 | A/G | 0.70 | 0.88 [0.84-0.94] | $1.1\times10^{-4}$ | 0.75 | 0.85 [0.82-0.88] | $1.1\times10^{-17}$ |
| | 11q13 | known | rs7927894 | 76,301,316 | C11orf30,LRRC32 | C/T | 0.34 | 1.08 [1.02-1.14] | $9.0\times10^{-3}$ | 0.37 | 1.10 [1.07-1.13] | $3.5\times10^{-11}$ |
| | **12q13** | new | rs167769 | 57,503,775 | STAT6,NAB2,LRP1 | C/T | 0.22 | 1.11 [1.04-1.19] | $6.9\times10^{-4}$ | 0.40 | 1.08 [1.05-1.11] | $1.6\times10^{-7}$ |
| | **15q22** | known | rs11071558 | 61,069,421 | RORA,NARG2, VPS13C | A/G | | | | 0.14 | 0.89 [0.85-0.92] | $1.9\times10^{-10}$ |
| | 15q22 | known | rs2033784 | 67,449,660 | SMAD3,SMAD6, AAGAB | A/G | 0.42 | 1.04 [0.99-1.10] | 0.23 | 0.30 | 1.11 [1.08-1.14] | $2.5\times10^{-14}$ |
| | **17q12-21** | known | rs2952156 | 37,876,835 | ERBB2,PGAP3, C17orf37 | A/G | 0.58 | 0.92 [0.87-0.97] | 0.01 | 0.70 | 0.86 [0.84-0.88] | $7.6\times10^{-29}$ |

CAAPA lead SNPs (MR-MEGA association $p < 10^{-6}$) and genome-wide significant associations reported by TAGC with evidence for replication in CAAPA are listed. Because some of the CAAPA studies were included in the TAGC asthma meta-analysis, and in order to contrast African and European asthma susceptibility loci, the table summarizes the associations in TAGC Europeans only (statistics from the TAGC random effects analysis, reported for the CAAPA and TAGC results). TAGC loci with strong evidence for association in CAAPA are highlighted in bold font. 1000 Genomes Project phase III European allele frequencies are reported as the estimated EAF for TAGC Europeans
*New* asthma GWAS result not reported prior to the corresponding GWAS, *Known* asthma GWAS result reported prior to the corresponding GWAS. *RA* reference allele, *EA* effect allele, *EAF* effect allele frequency, *P* P asthma association

correction for 810 association tests: chromosomes 9p24 (*RANBP6*, *IL33*), 15q22 (*RORA,NARG2,VPS13C*), and 17q12–q21 (Supplementary Table 13).

The TAGC lead SNP rs167769 on chromosome 12q13 is intronic to *STAT6*, a transcription factor that affects Th2 lymphocyte responses mediated by IL-4 and IL-13[7,22]. This was a new association reported by TAGC, not previously implicated in any asthma GWAS, although we note this SNP has been reported as a putatively causal SNP discovered by GWAS of lung function[23,24], and a number of linkage studies have pinpointed this chromosomal region in the early days of genome-wide investigations of asthma and atopy[25–28]. In addition, markers in *STAT6* have been identified by a number of candidate gene association studies[29–32]. However, prior to the CAAPA analyses, this locus had not been replicated in independent asthma GWAS datasets. The lead SNP rs3122929 in CAAPA at this locus is in strong LD with the TAGC lead SNP rs167769 in Europeans from the TGP ($r^2 = 0.93$) and nearly achieves genome-wide significance in CAAPA (MR-MEGA association $p = 9.1 \times 10^{-7}$). In addition, the association observed in TAGC subjects of African ancestry was marginal (fixed effect $p = 0.05$). With the increased sample size available through CAAPA, our meta-analysis provides further evidence of the association between the 12q13 region and asthma, confirming its contribution to asthma risk in African ancestry populations.

We also assessed windows ±10 KB from each of the 18 TAGC lead SNPs for replication in CAAPA (±200 KB for the chr17q12–q21 locus, due to the extended LD in this region), as well as 2 SNPs previously reported as achieving genome-wide significance in African ancestry populations[5,6] (Fig. 1c). Considering a Bonferroni-corrected significance threshold for 20 tests (one for each of these prior loci), an additional seven TAGC loci showed evidence of association in the CAAPA meta-analysis (Table 2, loci not in bold font). This includes a novel TAGC locus on chromosome 6p21 implicating human leukocyte antigen genes. In addition, for the known chromosome 5q31 TAGC locus, one of the CAAPA associations passing this significance threshold involved a SNP in strong LD with the TAGC lead SNP in TGP Europeans (rs1295686, $r^2$ with TAGC lead SNP rs20541 = 0.96, Supplementary Table 18).

The genome-wide meta-analysis of asthma in multi-ethnic populations previously performed by the EVE consortium reported a genome-wide significant African ancestry-specific association for SNP rs1101999, which is intronic to the *PYHIN1* gene[6]. This was the first African ancestry asthma association reported by a GWAS. This SNP has a high minor allele frequency in African populations (0.31 in the TGP phase 3 AFR population, $n = 1322$), but a very rare or low frequency in other populations (MAF < 0.005 in TGP phase 3 EAS, EUR, and SAS, $n = 1008$, 1006, and 978, respectively, MAF = 0.04 in AMR, $n = 694$), and the role this gene may play in asthma remains unclear. SNP rs1101999 was marginally associated with asthma in the CAAPA meta-analysis (MR-MEGA association $p = 6.4 \times 10^{-3}$), but the strength of association was much reduced from its original report (Supplementary Fig. 13, Supplementary Tables 12 and 14). Recently, the eMERGE (electronic medical records and genomics) network conducted an asthma GWAS in biobank subjects, and reported one genome-wide significant association in African Americans, SNP rs11788591, intronic to the *PTGES* gene. This association did not replicate in the CAAPA meta-analysis (MR-MEGA association $p = 0.23$, Supplementary Table 12, Fig. 1c, Supplementary Fig. 13).

**Genome-wide significant associations on chr17q12–q21.** Associations between SNPs on chr17q12–21 and risk of asthma showed evidence of ancestry heterogeneity (46 of the 54 SNPs with association $p < 10^{-6}$ had significant heterogeneity, $p < 0.05$). In general, the magnitude of the effect size increases as the average proportion of European ancestry in the study increased (Supplementary Table 9; most of the corresponding ß0 values were close to zero, whereas ß1, which captures the effect along the axis of genetic variation separating African and European ancestry, showed an increase in effect size magnitude as European ancestry increased). We also observe that the higher the average proportion of African ancestry, the smaller the magnitude of the effect size (Fig. 2b, forest plot of the lead association ordered by average African ancestry). An exception was observed among Honduran subjects who self-reported as Garifuna (HONDAS). HONDAS has a large average African (77%), a higher Native

**Table 3 Summary of tests for replication for two novel loci on chromosome 8 with p-values < 10⁻⁶ in the CAAPA meta-analysis**

| SNP | Ethnicity | Study | Number non-asthmatics | Number asthmatics | p-value | Effect direction |
|---|---|---|---|---|---|---|
| rs13277810 | **African ancestry** | **CAAPA** | **7009** | **7645** | **3.20E−08** | + |
|  | African American | CARDIA | 109 | 860 | 1.68E−01 | + |
|  | African American | MESA | 200 | 1437 | 5.44E−01 | − |
|  | African American | ARIC | 89 | 1636 | 5.24E−01 | − |
|  | African American | BioMe | 391 | 1550 | 1.07E−01 | − |
|  | Hispanic | BioMe | 519 | 1775 | 7.15E−04 | + |
| rs13269769[a] | European | TAGC | 19,954 | 107,715 | 9.55E−02[b] | − |
| rs114647118 | **African ancestry** | **CAAPA** | **7009** | **7645** | **2.70E−07** | − |
|  | African American | CARDIA | 109 | 862 | 5.61E−02 | + |
|  | African American | MESA | 200 | 1440 | 4.62E−01 | − |
|  | African American | ARIC | 89 | 1638 | 7.18E−01 | − |

rs13277810 is a common variant that reached genome-wide significance, and rs114647118 is a low frequency variant with MR-MEGA association p-value < 10⁻⁶. rs13277810 was not available in the TAGC meta-analysis summary statistics, but rs13269769, a SNP that has high LD with rs13277810 in the 1000 Genomes Project Europeans, was available. The rs114647118 association had low imputation quality in BioMe, and was not included in the TAGC meta-analysis summary statistic dataset. Effect direction is defined in terms of the minor allele. The CAAPA discovery associations are highlighted in bold font
[a]$r^2$ with rs13277810 in TGP EUR = 0.994
[b]Fixed-effect p-value

American (20%), and a very small European component (3%). Interestingly, the lead SNP in CAAPA was the same SNP reported by a meta-analysis of asthma in Puerto Rican children[33], distinct from the lead SNPs reported by the multi-ethnic EVE and TAGC GWAS[6,7]. The most significant and largest effect size magnitudes were observed for the studies that had higher European and Native American components; however, we speculate that this may reflect risk for asthma inherited from a Native American genetic background given the minimal European component in the HONDAS population. We note this trend is not as strong in PGCA, the CAAPA study with the highest proportion of Native American ancestry (29%), which may be due in part to the heterogeneous patterns of LD in this chromosomal region.

In addition to the smaller effect size magnitude in studies with high African ancestry, we also observed much weaker associations (Supplementary Fig. 11 vs. Supplementary Fig. 12), and no chr17q12–q21 associations with $p < 10^{-6}$ were observed in an inverse-variance meta-analysis of African American samples (Supplementary Table 4), despite a relatively large sample size of 3651 cases and 4222 controls. These observations are consistent with a recent report by Stein et al. summarizing the chromosome 17q12–q21 locus, which included an analysis and discussion of the relatively weak associations observed at this locus among African Americans[13]. They posited that the reduced strength of association may be due to an overall lower MAF spectrum in African Americans in this region (which reduces statistical power to detect association), breakdown of LD on African haplotypes, and different asthma endotypes (viral exposures) in children. Based on the reduced effect size magnitude observed in studies with higher average African ancestry, we additionally posit a smaller effect size magnitude is observed on African ancestry haplotypes (which could in turn be due to breakdown of LD and reduced correlation between tagging and causal variants). To investigate this, we extracted 17 putatively causal SNPs reported in the Stein et al. review, plus an additional five SNPs from the CAAPA meta-analysis with MR-MEGA association $p < 10^{-6}$ and $r^2 < 0.8$ between all 17 SNPs in TGP European and African populations. We then stratified subjects for whom local ancestry estimates were available based on the number of copies of African ancestry at each of these SNPs, and tested for association between each SNP and asthma separately for each local ancestry group (0, 1, or 2 copies of African ancestry, Supplementary Data 2). Figure 2c shows a trend of decreasing effect size magnitude as the number of copies of African ancestry increased, for most of these 22 candidate SNPs, suggesting (but not conclusively proving) smaller effect size magnitudes on African ancestry haplotypes.

Consistent with Stein et al., the degree to which LD breaks down in CAAPA samples increased with average African ancestry (Fig. 2a and Supplementary Fig. 18). One example of how this breakdown could affect association is the rs12936231–rs4065275 haplotype and its association with ORMDL3 expression. Specifically, Stein et al. notes that the rs12936231-C and rs4065275-G alleles were associated with high expression of ORMDL3 in peripheral blood cells from non-African populations, but the rs4065275-G allele was not associated with expression of ORMDL3 in Yoruban (African) lymphoblastoid cell lines. This is possibly due to the rs12936231-C and rs4065275-G haplotype almost always being present in this combination on non-African haplotypes, while the rs12936231-G and rs4065275-G haplotype in non-Africans was rare (3%). In contrast, the rs12936231-G and rs4065275-G haplotype is common in Africans (19%)[13]. We examined the rs12936231–rs4065275 haplotype for association with asthma in the three local ancestry groups (0, 1, or 2 copies of African ancestry), but no significant associations were observed in any of the groups (Supplementary Table 16).

Finally, we note genome-wide significant SNPs at the chromosome 17q12–q21 region in CAAPA ranged between positions 37,908,867–38,089,717 (Fig. 2a), which included the ORMDL3/GSDMB haplotype block (hg19, see Supplementary Table 9), at least 32 kb from the most significant SNP (at position 37,876,835) in TAGC. However, in the TAGC pediatric sub-group analysis the strongest association was observed 3.6 kb proximal to GSDMB, so we speculate the relatively strong association in CAAPA at the ORMDL3/GSDMB region reflects the large proportion of childhood onset asthmatics included in CAAPA (Supplementary Table 1). The ORMDL3/GSDMB region is strongly associated with childhood onset asthma[13,34], and to rule out that the weaker associations in high African ancestry populations may be due to the inclusion of adult onset asthmatics in some of the CAAPA studies, we re-tested the association between the 22 candidate SNPs and asthma, using only those studies where age of onset was available, or that were pediatric studies. We compared the association results when including versus excluding adult onset asthmatics, and the strength of the associations remained marginal when adult onset asthmatics were excluded (Supplementary Table 15).

**Associations with total serum IgE.** We also examined whether genetic associations with asthma overlap with atopy by testing lead SNPs from Table 2 for association with total serum IgE (tIgE) using 4132 subjects for whom this phenotype was available (CAAPA lead SNPs and lead SNPs from TAGC that replicated in

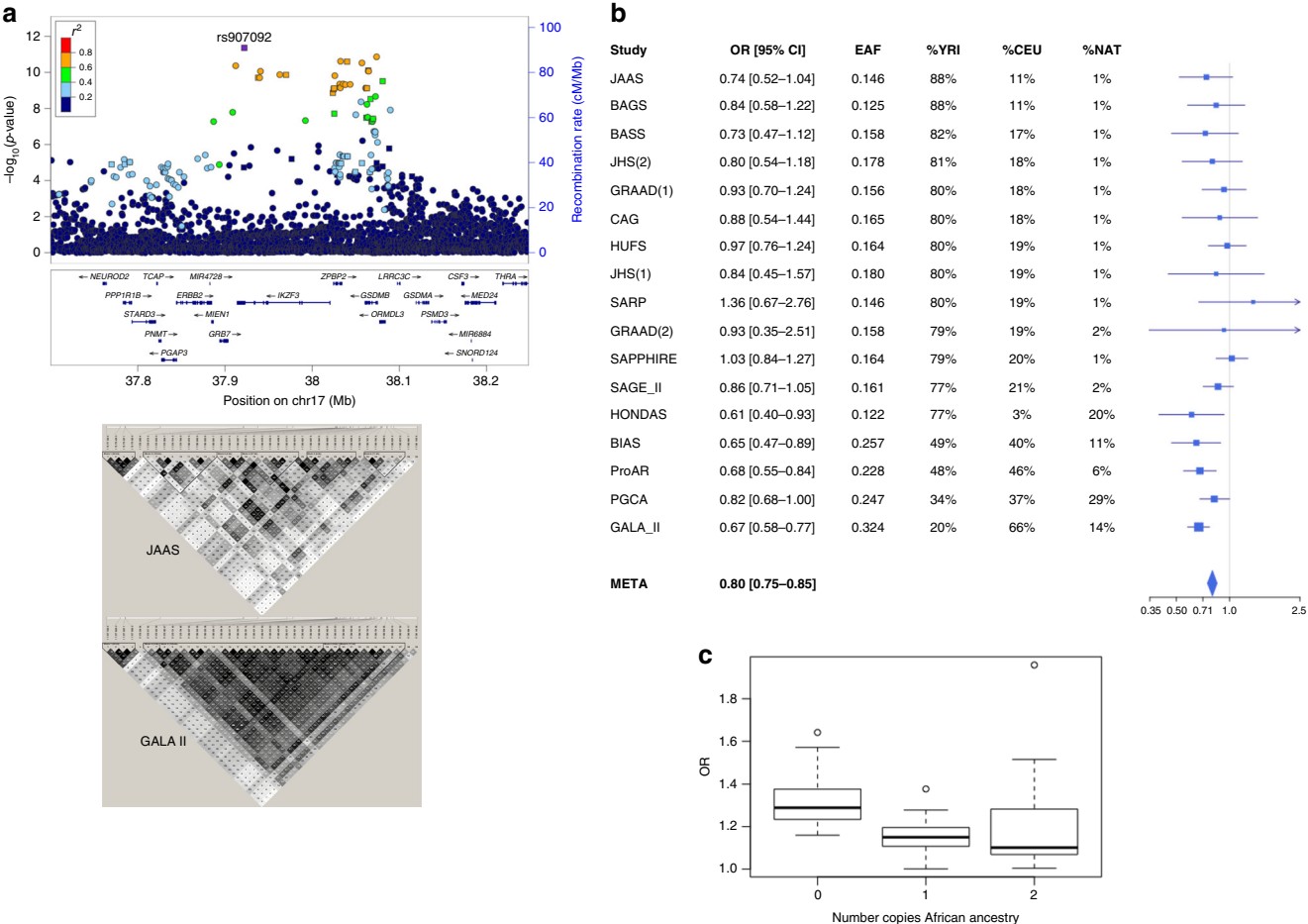

**Fig. 2** Summary of the chromosome 17q12–21 region results. **a** This panel shows a locus zoom plot of the CAAPA meta-analysis results. Positions of 39 SNPs associated with asthma in TAGC Europeans that replicated in CAAPA (from Supplementary Table 13) are denoted by squares. The $r^2$ between the lead SNP rs907092 and the other SNPs with associations (represented by red, orange, green, light, and dark blue symbols) was calculated using African American subjects from the CAAPA WGS reference panel. The $r^2$ between the 39 SNPs in JAAS and GALA II are shown at the bottom of the plot, with darker shades of grey representing higher LD. JAAS and GALA II are the studies with the highest and lowest proportions of African ancestry, respectively. **b** A Forest plot of the effect size of rs907092 is shown in this panel. CAAPA datasets are ordered by decreasing percentage African ancestry. EAF effect allele frequency. %YRI, CEU, and NAT represent estimated mean percentage of African, European, and Native American ancestry, respectively. **c** This panel shows a box plot of the asthma risk allele odds ratio for 22 candidate SNPs in the chromosome 17q12–21 region. The 17 SNPs discussed by Stein et al.[13] were selected, and an additional five SNPs from the CAAPA meta-analysis, with MR-MEGA association $p < 10^{-6}$ and $r^2 < 0.8$ with all 17 selected SNPs in the 1000 Genomes Project European and African populations, were also included. These five additional SNPs are also expression quantitative trait loci for *GSDMB/GSDMA/ORDLM3* in one or more GTEx tissues. The center line represents the median odds ratio, the box bounds represent the first and third quartile of the odds ratio distribution, and the whiskers are 1.5 times the first and third quartile odds ratio. Outliers are represented by circles

CAAPA; associations were tested separately in cases and controls and then combined using meta-analysis). Four of the SNPs correlated with levels of tIgE in asthmatics (with the asthma risk allele associated with increased levels of tIgE, Supplementary Table 19), suggesting that these asthma risk alleles may lead to an increased Th2 immune response. This includes SNP rs1420101 in the *IL1R1* locus (a gene that is known to correlate with eosinophilia[35]), SNP rs10519067 intronic to the *RORA* gene (encoding a transcription factor that regulates the growth of group 2 innate lymphoids, a key cell type in the memory Th2 cell response[36]), and SNPs rs907092 and rs2952156 in the chr17q12–21 locus. These results are consistent with previous studies that have shown both shared and unique associations between these phenotypes[37–41].

**Admixture mapping**. In addition to performing an asthma GWAS, we also leveraged local ancestry to identify regions of the genome where ancestry from a particular ancestral population

was inherited more frequently in affected versus unaffected individuals. This technique, called admixture mapping, can be used as a complementary approach to association mapping in admixed populations to uncover associations not detectable by SNP tests alone[42]. Because the genetic structure of the Barbados population is similar to that of African Americans (Fig. 1a and Supplementary Fig. 1), and because of overlap in genotype array coverage, we combined the African American studies genotyped on the ADPC and the Barbados study (BAGS) and used this combined dataset for admixture mapping discovery. CAAPA studies genotyped on the MEGA, additional BAGS subjects genotyped on Illumina's Omni array, and African American subjects from Bio*Me* were leveraged to replicate our admixture mapping discoveries. The distribution of *p*-values from the discovery admixture mapping tests for association is summarized in Supplementary Fig. 20 (local ancestry dose association *p*-values reported by EMMAX). The QQ plot shows little evidence of systematic test statistic inflation. The deflated inflation factor of 0.90 appears to be largely driven by $0.1 < p < 0.5$, whereas the

number of $p < 0.02$ is greater than expected, suggesting association results are enriched with local ancestry segments showing differences in ancestry between cases and controls. Only one segment of local ancestry, ranging from base pair positions 134,149,974–134,300,365 on chromosome 6, crossed the multiple testing threshold (see Methods). This segment falls within a local ancestry peak on chromosome 6q22.31–23.2, with increased African ancestry associated with increased risk of asthma (Supplementary Table 20). Genes falling in this segment include the Transcription Factor 21 (TCF21) and TATA-Box Binding Protein Like 1 (TBPL1) genes. The TCF21 gene has been shown to be differentially expressed in bronchial biopsies of asthmatics compared to controls[43]. The most significant SNP showing association in this region in the admixture mapping discovery group (rs111966851, Supplementary Fig. 6) has a MAF of 0.308 and 0.006 in Africans and Europeans, respectively ($n = 1322$ AFR and $n = 1006$ EUR TGP phase 3), which corroborates the idea that African ancestry in this segment may increase risk for asthma. For replication, local ancestry segments overlapping with the peak segments were tested for association with asthma in CAAPA studies with similar genetic structure to the discovery studies (i.e., African Americans and Jamaicans, see population structure in Fig. 1a and Supplementary Fig. 1), additional BAGS (Barbados) subjects, and African American subjects from BioMe. None of these segments were associated with asthma (Supplementary Table 21), and SNP rs111966851 (the most significant SNP in this region in the admixture mapping discovery) was also not associated with asthma in the CAAPA studies excluded from the admixture mapping discovery (Supplementary Fig. 16).

## Discussion

We report a large GWAS of asthma in African ancestry populations; prior studies included only 763–3037 asthmatic subjects[4–7]. Eleven of the 18 loci recently identified in the TAGC meta-analysis show evidence of association in CAAPA, including strong evidence for four different regions. This includes the region around STAT6 on chromosome 12q13, a novel region identified by TAGC and not replicated to date, as well as the well-known chromosome 17q12–21 region, which reached genome-wide significance in our analysis. It has been posited that disparities in asthma susceptibility can partly be explained by genetic risk factors[4,44,45]. In recapitulating associations mainly discovered in European ancestry populations (a result that has previously not been well-quantified in the literature), our results suggest that at the very least, common genetic variation may not strongly contribute to asthma disparities. However, our data show the chromosome 17q12–21 associations have smaller effect size magnitudes on African ancestry haplotypes. In addition, we built a genetic risk score for the 18 asthma loci reported by TAGC, and found that although asthmatics had a statistically significant higher risk score compared to controls (Supplementary Fig. 24), the effect was too small to build a predictive risk score for asthma in CAAPA.

In addition to recapitulating asthma genes discovered largely in non-African populations, we identified two loci on chromosome 8 not previously reported by asthma GWAS, and through admixture mapping identified a region on chromosome 6q22. In our inclusion of the largest sample of African ancestry individuals in our discovery GWAS, we were unfortunately limited in sample sizes of African ancestry individuals available for replication with consequent limitations in power. The most significant SNP on chromosome 8p23 reached genome-wide significance and was replicated in Hispanics from BioMe. However, our attempts to replicate this same locus in African Americans were unsuccessful; the number of cases compared to controls was considerably

smaller (4–18 times smaller) as the replication studies were not primarily ascertained for asthma. Similarly, we also failed to replicate the low frequency variant on chromosome 8q24 (only 398 African American asthmatics with genetic data were available) and the admixture mapping signal (only 845 cases were available, of which 498 were African American).

The CAAPA meta-analysis includes data from 15 independent studies and is the largest asthma GWAS focused on African ancestry populations to date. Unfortunately, as is the case for other complex diseases for which morbidity disproportionately affects underrepresented and underserved populations[46–48], a legacy of underrepresentation or exclusion of minorities from federally-funded studies has rendered comparatively robust non-European datasets rare to nonexistent. A recent analysis of ancestry represented in the GWAS Catalog[12] concluded that non-European, non-Asian groups combined account for less than 4% of individuals included in the catalog[46]. The analysis also found that African ancestry individuals contributed 7% of all catalog associations, despite only comprising 2.4% of the catalog, highlighting the value of GWAS conducted in African ancestry populations for enabling scientific discoveries. Furthermore, the authors stressed the importance of assessing the generalizability of genetic disease associations across populations, and the value that low-LD African ancestry individuals contribute to multi-ethnic fine-mapping of genetic associations. Despite the considerable federal support for the CAAPA initiative, we do recognize that the CAAPA sample size falls considerably short of the recent mega studies comprised of asthma datasets in the hundreds of thousands[7,49]. Furthermore, replication had to be sought in studies not primarily ascertained for asthma and with limited sample sizes. Important insights drawn from this study include the demonstration that many of the genetic loci associated with asthma in European ancestry populations may also be at play in African ancestry populations, and a clearer understanding of the LD patterns among African ancestry populations in 17q21. Potentially novel loci discovered by this meta-analysis are as yet not replicated within this study, but warrant follow-up. Importantly, the advent of institutional biobanks with access to multi-ethnic patient populations[50], as well as efforts by institutions such as the National Institute of Health to reduce health and research disparities[51] promise to greatly expand representation of well-characterized patients of African ancestry in the near future, allowing for robust follow-up of these CAAPA findings. The improved availability of African ancestry whole-genome sequence imputation reference panels available through initiatives such as the NHLBI-supported TOPMed program[52] should also provide high quality imputation of low and rare frequency variation in African ancestry populations, which will empower future studies. Lastly, we note better availability of other -omics datasets representing diverse ethnicities, such as transcriptomic data in tissue types relevant to asthma, will be needed to enable discoveries by utilizing the next generation of analysis tools[53–55].

## Methods

**Study oversight**. NIH guidelines for conducting human genetic research were followed. The Institutional Review Boards (IRB) of Johns Hopkins University (GRAAD, BASS and BAGS), Howard University (CRAD and HUFS), Wake Forest University (SARP), the University of California, San Francisco (coordinating center for the SAGE II and GALA II studies), the Western Institutional Review Board for the recruitment in Puerto Rico (GALA II Puerto Ricans), Children's Hospital and Research Center Oakland and Kaiser Permanente-Vallejo Medical Center (SAGE II), the University of Chicago (CAG), University of the West Indies, Mona, Jamaica and Cave Hill Campus, Barbados (BAGS), University of Mississippi Medical Center (JHS), Henry Ford Health System (SAPPHIRE), the Universidad Católica de Honduras in San Pedro Sula (HONDAS), Federal University of Bahia (BIAS and ProAR), the University of Cartagena (PGCA), all reviewed and approved this study. All participants provided written informed consent.

**Genome-wide ancestry estimation and analysis**. Unrelated phase 3 1000 Genomes Project (TGP) subjects of European ancestry (CEU, $n = 84$) and African ancestry (YRI, $n = 84$), as well as unrelated Native American (NAT) subjects from Mao et al.[56] ($n = 43$), were used as reference panels for the genome-wide ancestry analysis. A combined dataset of SNPs common to the reference panels plus all genotyped SNPs in the CAAPA datasets (with <1% missing genotypes in each dataset) were created, after which a LD pruning step was performed (leaving 20,482 SNPs). (Note for the SAPPHIRE dataset, the combined genome-wide ancestry analysis described here was restricted to the 730 asthmatic cases for which ADPC [African Diaspora Power Chip] data were available.) Principal components were then formed from the genotypes of the CAAPA and reference subjects by the R Bioconductor package GENESIS. GENESIS uses PC-AiR to calculate principal components to account for cryptic and known relatedness between subjects[57]. PC1 distinguished African ancestry from European and Native American ancestry and PC2 distinguished Native American ancestry from European and African ancestry. The results are summarized in Supplementary Fig. 1. The combined dataset was also used to estimate the proportion of genome-wide ancestry deriving from the three source populations represented by the reference panels, for each CAAPA subject. This was done using the software program ADMIXTURE. Because BAGS, HUFS, and BIAS include families, and ADMIXTURE assumes all subjects are unrelated, only the founders in these studies ($n = 226$, $n = 997$, and $n = 179$, respectively) were included for the ADMIXTURE analysis. These results are summarized in Fig. 1a and Supplementary Table 2. Finally, the GENESIS PCA was repeated separately for each of the CAAPA datasets, excluding the reference populations, using a set of LD pruned SNPs, in order to test for differences in ancestry between cases and controls within each dataset. The association between the first 2 PCs and asthma is summarized in Supplementary Table 2. As expected, the first principal component explained most of the variance in the genetic data (Supplementary Fig. 2) in all of the datasets. Relevant PCs were included in the asthma association models, in order to adjust for population structure (see Association analysis).

**Association analysis**. Because the studies included in the primary asthma association analysis represent a diverse spectrum of African ancestry (Fig. 1a and Supplementary Table 2), and had different objectives and differed in potentially relevant factors such as age of onset and diagnosis (Supplementary Note 1 and Supplementary Table 1), we performed tests for association separately for each dataset, and then combined the association results using MR-MEGA, a novel meta-regression approach that models allelic effects as a function of axes of genetic variation[11]. In this way, heterogeneous associations across genetically distinct populations are not penalized, and the degree of heterogeneity due to ancestry can be assessed, as well as residual effects that may be due to differences in study design.

**Statistical models fitted to the GWAS + ADPC datasets**. Logistic mixed effects models were used to test for association between imputed allelic dosage and asthma, using the GENESIS R Bioconductor package. GENESIS uses a penalized quasi-likelihood approximation to the generalized linear mixed model. SNP association $p$-values are estimated using a score test, which tests for model fit improvement if the SNP is added to the null model. GENESIS uses PC-relate to estimate a kinship matrix excluding other sources of variance such as population structure[58], and PC-AiR to calculate principal components accounting for cryptic and known relatedness between subjects[57]. The kinship matrix and principal components were calculated using a dataset of LD pruned genotyped SNPs. This kinship matrix was included as random effect in the null model, and principal components were included as fixed-effect covariates (the first principal component, as well as any of the top 10 principal components associated with asthma status [$p$-value < 0.05], were included).

**Statistical models fitted to the SAPPHIRE dataset**. ADPC data were only available for 730 SAPPHIRE asthmatics, but no controls. GWAS array data from the Affymetrix Axiom AFR array were available for 1325 cases and 566 controls (Table 1). The Henry Ford Health System group performed tests for association for this study, and shared summary statistics with CAAPA. To generate the summary statistics, logistic regression was used to test for association between imputed allelic dosage and asthma, using the PLINK software package[59,60]. The GENESIS R Bioconductor package was used to estimate a kinship matrix, to ensure that the coefficient of relatedness between each pair of subjects is below 0.25. Principal components were also calculated using GENESIS. Principal components included in the regression models were chosen using the same strategy used in the CAAPA GWAS + ADPC association analysis.

**Statistical models fitted to the MEGA datasets**. Asthma association tests in the MEGA datasets used the same statistical models described for the GWAS + ADPC datasets. However, because some of the populations genotyped on the MEGA have a broader ancestry spectrum (Fig. 1a, Supplementary Fig. 1), the association analysis pipeline included as fixed-effect covariates the first principal component, any other principal components identified by the elbow method (Supplementary Fig. 2)

as explaining a large percentage of variance, as well as any of the top 20 principal components associated with asthma status [$p$-value < 0.05].

**Processing and assessment of individual association results**. First, association results of SNPs with low imputation accuracy were removed from the individual datasets. The chosen filter was informed by a study that quantified imputation accuracy in African Americans[61] and filtering was done based on minor allele frequency (MAF) and the per-SNP estimation of the squared correlation between imputed allele dosages and true unknown genotypes (Rsq). Associations between asthma and SNPs with MAF ≤ 0.005 were excluded if Rsq ≤ 0.5, as were associations with SNPs with MAF > 0.005 if Rsq ≤ 0.3. SNPs with a minor allele count ≤10 were also excluded. Next, inflation factors and QQ plots were used to assess the individual study association results (Supplementary Figs. 3 and 4). Large inflation of these test statistics was not observed. All of the CAAPA datasets were therefore included in the meta-analysis, described below.

**Meta-analysis**. MR-MEGA uses multi-dimensional scaling to infer genetic axes of variation across studies, and models allelic effect across studies using a linear regression model of the allelic effect of each study along each genetic axis of variation, weighting the contribution of a study by the inverse variance of the allelic effect from the study. The model is described in detail in the Materials and Methods section of the MR-MEGA publication[11]. As input, the software requires the association odds ratio (OR) and its 95% confidence interval (CI), which were calculated per SNP and study as follows, where $U$ is the Score statistic of the model fitted by GENESIS for the SNP, and $i$ is its variance:

$$OR = e^{U/i} \qquad (1)$$

$$95\% \, CI = e^{U/i \pm 1.96\sqrt{1}/i} \qquad (2)$$

These formulas are based on recommendations from Zhou et al.[62]. For SAPPHIRE, the odds ratio reported by the software was used as is, and the 95% CI was calculated as follows, using the standard error (SE) reported by the software:

$$95\% \, CI = e^{\log(OR) \pm 1.96 SE} \qquad (3)$$

The MR-MEGA software requires specification of the number of genetic axes of variation to be used in the meta-analysis. Most of the CAAPA populations have ancestry from African and European populations, with some Native American ancestry (Fig. 1a). Because the mean Native American ancestry represented is small, however, and the first axis generally separates populations of high and low African ancestry (Supplementary Fig. 5), only 1 axis of genetic variation was used in the meta-analysis.

**Inverse-variance meta-analysis**. MR-MEGA estimates $p$-values by comparing the deviance of the regression model with coefficients equal to zero, to the deviance of the model with the coefficients unconstrained, and combined odds ratios/effect sizes across studies are not estimated. For this reason, all cross-study odds ratios reported in this paper, e.g., Table 2 and the forest plots in the supplementary material, were estimated using inverse-variance meta-analysis. Inverse-variance meta-analysis was also used to combine association results of all the African American studies in order to generate Supplementary Table 4, combined SNP association results for the admixture mapping peak in the admixture mapping discovery dataset (Supplementary Fig. 6), and assessing associations on chromosome 17q12–21 in studies with high and low African ancestry (Supplementary Figs. 11 and 12, Supplementary Data 2).

**Replication in cardiovascular disease studies**. Imputed GWAS array data (described previously[63]) from three cardiovascular disease studies with asthma information were used to assess replication of novel associations in CAAPA (CARDIA[64], MESA[65], and ARIC[66], see Supplementary Note 2 and Table 3). The imputed allelic dose of SNPs rs13277810 and rs114647118 was extracted, and logistic regression (implemented in the R software packages) was used to test for association between allelic dose and asthma. Principal components associated with asthma ($p$-value < 0.1) were included as covariates in the models.

**Replication in BioMe**. Summary statistics of tests for association between GWAS array data and asthma in African Americans and Hispanics from BioMe were also used to assess replication (see Supplementary Note 2 and Table 3). These summary statistics were generated using the logistic regression implementation in PLINK[59]. ImpG-Summary[67] and the CAAPA WGS reference panel (Supplementary Table 3, Supplementary Note 3) were used to impute associations surrounding rs13277810, separately for African Americans and Hispanics (predicted $r^2$ for rs13277810 was high, 0.911 in both the African American and Hispanic datasets). Associations for the region surrounding the low frequency SNP rs114647118 were also imputed, but the imputation quality of this SNP, as well as other low frequency SNPs in high LD

with it, was insufficient to assess replication (predicted $r^2 = 0.586$ and 0.581 for all SNPs assessed, in the African American and Hispanic datasets, respectively).

**TAGC SNP-by-SNP comparison**. The CAAPA meta-analysis was compared to the TAGC multi-ethnic meta-analysis, the largest and most definitive asthma GWAS to date[7]. The association results of the lead SNPs reported by TAGC were compared by extracting the corresponding CAAPA meta-analysis p-values, and using inverse-variance meta-analysis to calculate odds ratios and 95% CI for all SNPs (Table 2 and Supplementary Table 12). Note that two of the TAGC lead SNPs were not present in CAAPA: rs2855812 was filtered out of the CAAPA WGS reference panel because this SNP is located in a segmental duplication region, and rs11071558 was filtered out from the CAAPA reference panel because it is tri-allelic. Because the largest population represented in TAGC is European (19,954 of the 23,948 cases and 107,715 of the 118,538 controls), and because some of the CAAPA studies were included in the TAGC multi-ethnic meta-analysis, a direct comparison of association significance and effect size was only done for the TAGC European meta-analysis, and not the multi-ethnic meta-analysis. Together with the ancestry heterogeneity p-value estimated by MR-MEGA, this is also useful for gleaning the effect of asthma-associated SNPs in Europeans versus Africans.

**TAGC replication in CAAPA**. In addition to the SNP-by-SNP comparison described above, the TAGC summary statistics were downloaded and merged with the CAAPA meta-analysis results. Associations of all SNPs with significant fixed-effect p-values in TAGC Europeans were assessed in the CAAPA meta-analysis, after applying a Bonferroni correction for the number of overlapping SNPs in the merged dataset. A Bonferroni correction for the number of SNPs assessed in CAAPA was applied as a stringent correction for claiming robust replication of loci reported by TAGC in CAAPA (Supplementary Table 13).

**Additional replication in CAAPA**. Windows ±10 KB from each of the 18 TAGC lead SNPs were extracted from the CAAPA meta-analysis results (±200 KB for the chr17q12–21 locus, due to the extended LD in this region). A ±10-KB window surrounding two SNPs reported as genome-wide significant in African ancestry GWAS was also extracted[5,6]. A Bonferroni-corrected significance threshold for 20 tests (one for each of these prior loci) was used as a measure of additional evidence of replication in the CAAPA meta-analysis (Fig. 1c and Table 2, loci not in bold font).

**LD block replication in CAAPA**. SNPs within the same LD block and in high LD with the TAGC lead SNPs ($r^2 > 0.8$) in Europeans from the TGP were also selected for comparison, and their association results were inspected (Supplementary Table 18). In this way, no or marginal replication of TAGC lead SNPs in CAAPA due to differences in LD patterns between Europeans and Africans can be assessed. The LD block surrounding each lead TAGC SNP was identified using TGP phase 3 data, population EUR, representing European ancestry. Thus, the effect of SNPs that may be causal in Europeans but not the lead SNP reported in the TAGC European analysis can still be tested for replication in CAAPA. Flanking SNPs ±10 KB from each TAGC lead SNP with MAF >0.05 in the CAAPA WGS reference panel and that intersected with the CAAPA meta-analysis association results as well as TGP phase 3 variants were selected, and Gabriel's algorithm implemented in Haploview[68,69] was then used to identify LD blocks present in the window of SNPs. CAAPA meta-analysis p-values for SNPs falling in the LD block containing the lead SNP, and that had $r^2 > 0.8$ with the lead SNP, were extracted, and these SNPs and their asthma associations are summarized in Supplementary Table 18. The chr17q12–21 locus achieved genome-wide significance in CAAPA and was therefore excluded from this analysis.

**EVE replication sensitivity analysis**. A sensitivity analysis of the African ancestry-specific genome-wide association reported by the EVE consortium for SNP rs1101999[6] was also done (Supplementary Table 12). CAAPA studies included in the EVE meta-analysis were meta-analyzed together, and all CAAPA studies not included in the EVE meta-analysis were meta-analyzed together. MR-MEGA was used for this meta-analysis, as CAAPA studies with relatively high proportions of European ancestry could be included, without penalizing the strength of the association should these high European ancestry populations show a different pattern of association compared to populations with high African and low European ancestry.

**Tests for association with total serum IgE**. Lead SNPs from Table 2 (reported by CAAPA, as well as TAGC SNPs with evidence for replication in CAAPA) were tested for association with total serum IgE (tIgE), adopting the approach used in the EVE consortium's meta-analysis of genetic association with tIgE[40]. The analysis was stratified based on asthma case-control status and originating study (eight groups of asthmatics and six groups of controls, see Supplementary Table 1 for a summary of the distribution of tIgE in these groups). Linear mixed-effect models implemented in the R Bioconductor GENESIS software package were used for association testing between the allelic dose at each SNP and Studentized residuals of log10 transformed tIgE (adjusted for age and sex). Similar to the asthma

association models, the models included a kinship matrix as random effect and principal components as fixed-effect covariates. Finally, association test statistics from the analysis strata were combined using inverse-variance meta-analysis, to yield combined statistics for non-asthmatics, asthmatics, and non-asthmatics + asthmatics (Supplementary Table 19).

**Admixture mapping discovery**. Local ancestry inference was performed using RFMix[70], and the pipeline is described in Supplementary Note 12. Custom scripts were used to convert the RFMix local ancestry calls to an EMMAX dosage TPED file, where the encoded dosage is defined as having 0, 1, or 2 copies of African ancestry at a particular local ancestry segment. Tests for association between the local ancestry dosage values and asthma case-control status were then done using the linear mixed-effect models implemented in the EMMAX software package. Phenotypes for subjects from SAPPHIRE, which included only asthmatics and no controls as well as the three population outliers identified by the genome-wide ancestry analysis, were set to missing. These models included dataset as a fixed-effect covariate, and a Balding-Nichols kinship matrix as random effect. The method described by Gao et al. was used to estimate the number of effective tests that should be used in a Bonferroni correction for multiple testing[71]. Briefly, a local ancestry correlation matrix, and corresponding eigenvalues, was calculated using local ancestry dosage values (0/1/2 copies of African ancestry) per chromosome. The number of effective tests for a chromosome was then set to the number of eigenvectors that explain 99.5% of the variance in the local ancestry data (for $n$ local ancestry segments, find the largest $k$ such that $\sum_{i=1}^{k}$ eigenvalues / $\sum_{i=1}^{n}$ eigenvalues ≤ 0.995). The total number of effective tests was then set to the sum of the number of effective tests calculated for each of the 22 autosomes. Using this method, a Bonferroni-corrected p-value threshold of 0.05/ (262 total number of effective tests) = $1.9 \times 10^{-4}$ was used to claim statistical significance.

**Admixture mapping replication**. The segments including the start and end position of the admixture mapping discovery peak (chr6:134,149,974–134,300,365) were selected for replication. Logistic regression was used to test for association between the number of copies of African ancestry (dosage value of 0, 1, 2) and case-control status, separately for each replication dataset (BASS, BAGS, JAAS, and BioMe). The base R package was used to fit the model to the BioMe data, adjusting for the first and fifth principal components, as these principal components were also associated with asthma status (p-values < 0.1). The R Bioconductor package GENESIS was used to fit models to the BASS, BAGS, and JAAS dataset, including a kinship matrix as random effect and principal components as fixed effects, as described for the SNP association analysis. The results for the segment including/ closest to the midpoint of the admixture mapping discovery peak (134,225,170) was combined using inverse-variance meta-analysis.

**Admixture mapping power calculations**. We used the Genetic Association Study (GAS) Power Calculator[72] to perform post-hoc power calculations for admixture mapping. As BAGS and HUFS include related subjects, their effective number of cases and controls were estimated as $n_{cases} = 1/(1 + 2\bar{r})$ and $n_{controls} = 1/(1 + 2\bar{r})$, where $\bar{r}$ is the mean kinship coefficient between relatives in the particular study[73] (the kinship matrix estimated by GENESIS (see Statistical models fitted to the GWAS + ADPC datasets) was used to calculate $\bar{r}$; $\bar{r} = 0.30$ for BAGS, $\bar{r} = 0.33$ for HUFS). Given 2380 cases and 3255 controls, at a significance level of $p = 1.9 \times 10^{-4}$, disease prevalence of 0.1, disease allele frequency of 0.2 (the mean number of copies of non-African ancestry), we had ≥89% power to detect a relative risk ≥1.20 in our discovery dataset. For our replication dataset, given 777 cases and 2263 controls, at a significance level of $p = 0.05$, disease allele frequency of 0.15 (accounting for the lower mean number of copies of non-African ancestry in BAGS and JAAS), we had ≥70% power to detect a relative risk ≥1.2. However, for a relative risk of 1.12 (the ratio of non-European local ancestry in controls versus cases for the segment identified in the discovery), we only had 34% power for replication.

**Software**. Detailed information on the software packages and software versions used for analyses are listed in Supplementary Table 24.

## Data availability

The legacy GWAS array ADPC and MEGA data that support the findings of this study have been deposited in dbGAP with the accession code phs001123.v2.p1. These data can be accessed through dbGAP. Specific data use limitations: GRU-IRB (General Research Use, IRB approval required). Only ADPC data are available for the SAGE II, GALA II, and SAPPHIRE datasets; in addition, no phenotype data are available for these three datasets. Summary statistics from the meta-analysis are also available through the GWAS catalog[12] [https://www.ebi.ac.uk/gwas/downloads/summary-statistics].

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

## Acknowledgements

We thank Goncalo Abecasis for coordinating inclusion of the CAAPA reference panel on the Michigan Imputation Server, Todd Deppe, Estelle Giraud, Cindy Lawley from Illumina for genotyping services, and Pat Oldewurtel for administrative and technical support.

## Author contributions

M.D., N.R., and T.M.B, analysed the data, interpreted the data, and wrote the paper. S.C., A.M.L., A.S., M.P.B., G.W., H.R.J. and G.B. analysed and interpreted the data. M.C. performed genotyping. K.C.B., R.A.M., T.H.B. and M.A.T. conceived the experiments, interpreted the data, and wrote the paper. C.V., C.R.G., V.E.O., A.D., D.G.T., N.A., M.I.A., P.C.A., E.B., C.B., L.C., A.C., G.M.D., C.E., M.U.F., T.S.F., C.F., J.G.F., W.G., P.A.G., N.N.H., R.D.H., E.F.H., S.J., E.E.K., J.K., R.K., L.A.L., E.M.L., A.L., P.M., T.M., A.M., D.M., D.L.N., T.D.O., R.R.O., C.O.O., O.O., Z.S.Q., C.R., N.V., H.W., R.J.W., J.G.W., S.S., C.O., E.G.B., L.K.W. and I.R. contributed to interpretation of results and critically reviewed the manuscript. CAAPA provided infrastructure, biospecimens and associated phenotype data, and intellectual input for the overall design and execution of the study.

## Additional information

**Competing interests:** All authors declare no competing interests.

Michelle Daya[1], Nicholas Rafaels[1], Tonya M. Brunetti[1], Sameer Chavan[1], Albert M. Levin[2], Aniket Shetty[1], Christopher R. Gignoux[1], Meher Preethi Boorgula[1], Genevieve Wojcik[3], Monica Campbell[1], Candelaria Vergara[4], Dara G. Torgerson[5], Victor E. Ortega[6], Ayo Doumatey[7], Henry Richard Johnston[8], Nathalie Acevedo[9], Maria Ilma Araujo[10], Pedro C. Avila[11], Gillian Belbin[12], Eugene Bleecker[13], Carlos Bustamante[3], Luis Caraballo[9], Alvaro Cruz[14], Georgia M. Dunston[15], Celeste Eng[5], Mezbah U. Faruque[16], Trevor S. Ferguson[17], Camila Figueiredo[18], Jean G. Ford[19], Weiniu Gan[20], Pierre-Antoine Gourraud[21], Nadia N. Hansel[4], Ryan D. Hernandez[22], Edwin Francisco Herrera-Paz[23,24], Silvia Jiménez[9], Eimear E. Kenny[12], Jennifer Knight-Madden[17], Rajesh Kumar[25], Leslie A. Lange[1], Ethan M. Lange[1], Antoine Lizee[21], Pissamai Maul[26], Trevor Maul[26], Alvaro Mayorga[27], Deborah Meyers[13], Dan L. Nicolae[28], Timothy D. O'Connor[29], Ricardo Riccio Oliveira[30], Christopher O. Olopade[31], Olufunmilayo Olopade[28], Zhaohui S. Qin[32], Charles Rotimi[7], Nicolas Vince[21], Harold Watson[33], Rainford J. Wilks[17], James G. Wilson[34], Steven Salzberg[35], Carole Ober[36], Esteban G. Burchard[22], L. Keoki Williams[37], Terri H. Beaty[38], Margaret A. Taub[39], Ingo Ruczinski[39], CAAPA, Rasika A. Mathias[4] & Kathleen C. Barnes[1]

[1]Department of Medicine, University of Colorado Denver, Aurora, CO 80045, USA. [2]Department of Public Health Sciences, Henry Ford Health System, Detroit, MI 48202, USA. [3]Department of Genetics, Stanford University School of Medicine, Stanford, CA 94305, USA. [4]Department of Medicine, Johns Hopkins University, Baltimore, MD 21224, USA. [5]Department of Medicine, University of California San Francisco, San Francisco, CA 94143, USA. [6]Center for Human Genomics and Personalized Medicine, Wake Forest School of Medicine, Winston-Salem 27157, USA. [7]Center for Research on Genomics & Global Health, National Institutes of Health, Bethesda, MD 20892, USA. [8]Department of Human Genetics, Emory University, Atlanta, GA 30322, USA. [9]Institute for Immunological Research, Universidad de Cartagena, Cartagena 130000, Colombia.

[10]Immunology Service, Universidade Federal da Bahia, Salvador 401110170, Brazil. [11]Department of Medicine, Northwestern University, Chicago, IL 60611, USA. [12]Department of Genetics and Genomics, Icahn School of Medicine at Mount Sinai, New York, NY 10029, USA. [13]Department of Medicine, University of Arizona College of Medicine, Tucson, AZ 85724, USA. [14]Universidade Federal da Bahia, Salvador 401110170, Brazil. [15]Department of Microbiology, Howard University College of Medicine, Washington, DC 20059, USA. [16]National Human Genome Center, Howard University College of Medicine, Washington, DC 20059, USA. [17]Caribbean Institute for Health Research, The University of the West Indies, Kingston 00007, Jamaica. [18]Departamento de Biorregulacao, Universidade Federal da Bahia, Salvador 401110170, Brazil. [19]Department of Medicine, Einstein Medical Center, Philadelphia, PA 19141, USA. [20]National Heart, Lung and Blood Institute, National Institutes of Health, Bethesda, MD 20892, USA. [21]Université de Nantes, INSERM, Centre de Recherche en Transplantation et Immunologie, UMR, 1064ATIP-Avenir, Equipe 5, Nantes, France. [22]Department of Bioengineering and Therapeutic Sciences, University of California San Francisco, San Francisco, CA 94143, USA. [23]Facultad de Medicina, Universidad Católica de Honduras, San Pedro Sula 21102, Honduras. [24]Universidad Tecnológica Centroamericana (UNITEC), Facultad de Ciencias Médicas, Tegucigalpa, Honduras. [25]Department of Pediatrics, Northwestern University, Chicago, IL 60611, USA. [26]Genetics and Epidemiology of Asthma in Barbados, The University of the West Indies, Chronic Disease Research Centre, Jemmots Lane, St. Michael BB11115, Barbados. [27]Centro de Neumologia y Alergias, San Pedro Sula 21102, Honduras. [28]Department of Medicine, University of Chicago, Chicago, IL 60637, USA. [29]Institute for Genome Sciences, University of Maryland School of Medicine, Baltimore, MD 21201, USA. [30]Laboratório de Patologia Experimental, Centro de Pesquisas Gonçalo Moniz, Salvador 40296-710, Brazil. [31]Department of Medicine and Center for Global Health, University of Chicago, Chicago, IL 60637, USA. [32]Department of Biostatistics and Bioinformatics, Emory University, Atlanta, GA 30322, USA. [33]Faculty of Medical Sciences, The University of the West Indies, Queen Elizabeth Hospital, Bridgetown, St. Michael BB11000, Barbados. [34]Department of Physiology and Biophysics, University of Mississippi Medical Center, Jackson, MS 39216, USA. [35]Departments of Biomedical Engineering and Biostatistics, Johns Hopkins University, Baltimore, MD 21205, USA. [36]Department of Human Genetics, University of Chicago, Chicago, IL 60637, USA. [37]Center for Individualized and Genomic Medicine Research, Henry Ford Health System, Detroit, MI 48202, USA. [38]Department of Epidemiology, Bloomberg School of Public Health, JHU, Baltimore, MD 21205, USA. [39]Department of Biostatistics, Bloomberg School of Public Health, JHU, Baltimore, MD 21205, USA. These authors contributed equally: Rasika A. Mathias, Kathleen C. Barnes.

## CAAPA

Ayola Akim Adegnika[40], Ganiyu Arinola[41], Ulysse Ateba-Ngoa[40], Gerardo Ayestas[23], Hrafnhildur Bjarnadóttir[42], Adolfo Correa [43], Said Omar Leiva Erazo[23], Marilyn G. Foreman[44], Cassandra Foster[4], Li Gao[4], Jingjing Gao[45], Leslie Grammer[11], Mark Hansen[46], Tina Hartert[47], Yijuan Hu[32], Iain Königsberg[1], Kwang-Youn A. Kim [48], Pamela Landaverde-Torres[23], Javier Marrugo[49], Beatriz Martinez[49], Rosella Martinez[23], Luis F. Mayorga[23], Delmy-Aracely Mejia-Mejia[50], Catherine Meza[49], Solomon Musani[43], Shaila Musharoff[3], Oluwafemi Oluwole[28], Maria Pino-Yanes [5], Hector Ramos[23], Allan Saenz[23], Maureen Samms-Vaughan[51], Robert Schleimer[11], Alan F. Scott[52], Suyash S. Shringarpure[3], Wei Song[29], Zachary A. Szpiech [22], Raul Torres [53], Gloria Varela[23], Olga Marina Vasquez[54], Francisco M. De La Vega[3], Lorraine B. Ware[47] & Maria Yazdanbakhsh [55]

[40]Centre de Recherches Médicales de Lambaréné, BP:242, Lambaréné 13901, Gabon. [41]Department of Chemical Pathology, University of Ibadan, Ibadan 900001, Nigeria. [42]Faculty of Medicine, University of Iceland, 101 Reykjavík, Iceland. [43]Department of Medicine, University of Mississippi Medical Center, Jackson, MS 39216, USA. [44]Pulmonary and Critical Care Medicine, Morehouse School of Medicine, Atlanta, GA 30310, USA. [45]Data and Statistical Sciences, AbbVie, North Chicago, IL 60064, USA. [46]Illumina, Inc., San Diego, CA 92122, USA. [47]Department of Medicine, Vanderbilt University, Nashville, TN 37232, USA. [48]Department of Preventive Medicine, Northwestern University, Chicago, IL 60611, USA. [49]Instituto de Investigaciones Immunologicas, Universidad de Cartagena, Cartagena 130000, Colombia. [50]Facultad de Ciencias de la Salud, Universidad Tecnológica Centroamericana (UNITEC), San Pedro Sula 21102, Honduras. [51]Department of Child Health, The University of the West Indies, Kingston 00007, Jamaica. [52]Department of Medicine, Johns Hopkins University, Baltimore, MD 21287, USA. [53]Biomedical Sciences Graduate Program, University of California San Francisco, San Francisco, CA 94158, USA. [54]Centro Medico de la Familia, San Pedro Sula 21102, Honduras. [55]Department of Parasitology, Leiden University Medical Center, Leiden 02333, Netherlands

