## [Peer Review File · Nature Communications]

Reviewer #1 (Remarks to the Author):

Summary

Daya et al present a GWAS of asthma among of African-admixed populations, consisting of 7009 cases and 7645 controls from many sites in America. This well written paper describes careful and appropriate analyses for the largest group of African Ancestry subjects with asthma presented to date. While the sample size represents a substantial effort by multiple groups throughout America, it does not have sufficiently high statistical power to detect associations, especially considering the heterogeneity of study populations (in terms of genetic ancestry, exposures, asthma-related outcomes and measures) that compose the meta-analysis. While some of the results presented are interesting, the manuscript is largely descriptive and does not provide strong evidence for any given result.

Major

- 1) Authors should be commended for their effort to increase diversity of asthma genetics studies, but as they recognize, they still have low power to detect associations: TAGC was composed of 140K subjects (7000 African ancestry) vs. CAAPA 14,000 persons with African ancestry. An improvement in sample size, but still an order magnitude less than that of European ancestry studies. As such, the current results, particularly those where CAAPA is the discovery cohort, are speculative without further validation.
- 2) The rationale for using multiple comparisons correction statistical thresholds is reasonable as presented, but the need to employ different thresholds for different approaches is consistent with there not being strong and reliable association signals. For example, the EVE PYHIN1 association that was identified with some of the same cohorts that are part of CAAPA does not replicate in CAAPA. This suggests that a future CAAPA study with a larger sample size will find different associations than those listed here.
- 3) The 17q21 region results are interesting, and while asthma associations at this locus are not novel, understanding LD patterns among populations of different ancestry may help link specific variants to asthma. In addition to focusing on the SNPs listed in the Ober summary of this region, a closer comparison of the genome-wide significant SNPs in this region in CAAPA vs. those reported previously by TAGC would be of interest. Can TAGC results be presented along with those of Fig 2A? Figures that relate association results to the haplotype blocks among groups with more/less African ancestry as shown in Fig 2D and Fig S18.
- 4) Until there are larger numbers of subjects of African Ancestry, comparing measures of association across racial/ethnic groups remains premature, but nonetheless, the finding that many of the European ancestry GWAS results are also observed in African Ancestry cohorts suggests that common genetic variation may not strongly contribute to asthma disparities. A discussion about the main message conveyed by title is lacking in the Discussion section.

Minor

- 1) Suggested format change of P-values listed in Table 1 to not using scientific notation for $p > 10^{-3}$

Reviewer #2 (Remarks to the Author):

This paper describes the largest asthma GWAS to date in African ancestry populations. The paper is generally well written and conclusions are appropriate for the data presented. The following issues require attention.

- 1 Although this is the largest GWAS in African ancestry populations looking at asthma to date, the study size still remains modest, which probably explains the relative lack of either known or novel genome wide significant signals identified. Some known signals were identified including the well described chr17 locus, although only 2 reached conventional genome wide significance (17q and

8p23, although attempts to replicate the latter in smaller African ancestry populations were not successful). The results are hence rather incremental although still worthwhile to report.

2 I note the degree of African ancestry varied quite widely between the populations included. To deal with this the results were adjusted for the spectrum of ancestry in the meta-analysis. I was not entirely clear how this was done but reassurance that this approach did not potentially weaken signals seen in the populations with the highest extent of African ancestry would be useful. It also appears that different chips were used for different populations and the imputed genotype results then meta-analysed: the imputation data used for this came from WGS data in relevant populations although again the degree of African ancestry varies somewhat in these populations.

3 Whilst some data on the subjects included for study have been previously reported it would probably help readers of the manuscript if a brief summary of inclusion criteria for cases and controls was included in the supplementary methods (how was asthma defined?; are the controls age matched?).

4 I assume additional phenotype data are also available eg on atopy and eosinophil counts: as some of the genetic associations between these phenotypes (eg eosinophil count and the IL1RL1 locus) overlap with asthma have these also been examined?

5 I note that an attempt to look at known associations has been made mostly on a SNP by SNP basis. What would be most interesting is to try and look at whether the genetic landscape of asthma is similar in African ancestry individuals or actually quite different. Whilst the study lacks much power to look at this, it should be possible to use TAGC data to generate an asthma risk score using multiple SNPs having genome wide significant effects, and then to look at how this risk score predicts asthma in the African ancestry population. One minor issue here is that some of the individuals used in the current analysis were included in TAGC analyses although they form a small proportion of the 142k individuals in TAGC.

6 It is suggested that the potentially novel (but not replicated) 8p23 signal could be driven by variants in either ARHGEF10 or MYOM2: a little more detail on how the credible set of variants was identified for eQTL look ups and what expression data were then used to define a credible set of genes would be helpful.

Reviewer #3 (Remarks to the Author):

The paper by Daya et al describes the largest genome-wide meta-analysis for asthma in African ancestry individuals. Undertaking such studies in non-European ancestry groups is going to be crucial in bringing genetically-driven medical advances to these populations, and the authors should be commended for the scale and ambition of the study as well as for the rigour with which the study was undertaken. Whilst the study yields little in the way of novel signals meeting the most stringent genome-wide levels of significance and with convincing replication, it nevertheless yields important insights and it undoubtedly represents a major step from studies of this kind previously undertaken. The admixture mapping adds a novel element. It will be of interest to the scientific community, especially those with particular interests in genetics understanding of disease in non-European ancestry populations or in respiratory health and disease.

Major comments

1. Replication:

a. Of the two new loci on Chromosome 8, the evidence from replication studies is presented in Suppl Table 14. This should be moved to the main paper so that the reader can evaluate the evidence for themselves.

b. Results from BioMe do support the first of these signals (but were unavailable for the second). One feature not discussed is that CARDIA, MESA and ARIC differ quite fundamentally from the discovery studies in having a much larger number of asthma cases than controls (18 times as many cases for ARIC), whereas the discovery cohorts had more controls than cases. Might the results in the replication cohorts be subject to ascertainment bias?

c. The results from these studies do not support replication but these analyses would be

underpowered compared to BioMe, for example. My interpretation of the analyses would be that the 8p23 signal is supported and that adequate replication for the 8p24 signal is still required. This could be covered more clearly in the discussion.

d. I am not concerned about the results being reported without further replication – the limitation here is the availability of relevant datasets and the authors should be commended for the analysis.

2. The reader would benefit from greater assistance to navigate through the large amount of information in the main text and supplement and improved connectivity between elements. For example, where cohorts are referred to in the main text by geographical origin e.g. lines 322-323, or by cohort name only (Figure 1A) it would be easier if the two were put together. Perhaps the information on location from Table S1 could be moved to the main text and figure 1A annotated by geographical origin. It would make it much easier to see at a glance which cohorts included African American samples.

3. Whilst it is reasonable that some of the cohorts have been included in a previous meta-analysis (TAGC), it would be helpful for the reader to see (ideally in the main text) what the overlap between TAGC and the current (CAAPA) analysis. The authors do appear to have removed overlapping studies when making key comparisons.

Minor comments

Line 43 – show (rather than “shows”)

Novel line 104 – novel in which study

Line 105 – missing “8” after chromosome?

Line 128 – omit “critical” (I don’t see that this threshold is critical)

Line 139 –missing “gene encoding”?

Line 209-221: Is the Chromosome 12 signal the same as one previously described for lung function (FEV1/FVC, Soler Artigas et al Nat Genet 2011) at LRP1? STAT6 (mentioned on line 213) was described as one of the “plausible genes” for this locus in Wain et al Nat Genet 2017 (suppl Table 15).

Perhaps the authors can comment in the methods or discussion on the power for admixture mapping in discovery and replication.

We thank the reviewers for their helpful comments. We have revised our manuscript accordingly. Our response to each point raised by the reviewers is in *blue italic font* below, and excerpts from the updated manuscript are underlined.

Reviewer #1 (Remarks to the Author):

Summary

Daya et al present a GWAS of asthma among of African-admixed populations, consisting of 7009 cases and 7645 controls from many sites in America. This well written paper describes careful and appropriate analyses for the largest group of African Ancestry subjects with asthma presented to date. While the sample size represents a substantial effort by multiple groups throughout America, it does not have sufficiently high statistical power to detect associations, especially considering the heterogeneity of study populations (in terms of genetic ancestry, exposures, asthma-related outcomes and measures) that compose the meta-analysis. While some of the results presented are interesting, the manuscript is largely descriptive and does not provide strong evidence for any given result.

Major

1) Authors should be commended for their effort to increase diversity of asthma genetics studies, but as they recognize, they still have low power to detect associations: TAGC was composed of 140K subjects (7000 African ancestry) vs. CAAPA 14,000 persons with African ancestry. An improvement in sample size, but still an order magnitude less than that of European ancestry studies.

We understand the reviewer's sentiments, and we accept that, compared to other large discovery samples, CAAPA is relatively small. We wish to note that 48% of the 7,000 African ancestry subjects represented in TAGC are included in CAAPA, and most importantly, the TAGC results are based on >90% European ancestry samples.

We now present the following arguments in the updated last paragraph of the Discussion: The CAAPA meta-analysis includes data from 15 independent studies and is the largest asthma GWAS focused on African ancestry populations to date. Unfortunately, as is the case for other complex diseases for which morbidity disproportionately affects underrepresented and underserved populations(1-3), a legacy of underrepresentation or exclusion of minorities from federally-funded studies has rendered comparatively robust non-European datasets rare to nonexistent. A recent analysis of ancestry represented in the GWAS Catalog(4) concluded that non-European, non-Asian groups combined account for less than 4% of individuals included in the catalog(1). The analysis also found that African ancestry individuals contributed 7% of all catalog associations, despite only comprising 2.4% of the catalog, highlighting the value of GWAS conducted in African ancestry populations for enabling scientific discoveries. Furthermore, the authors stressed the importance of assessing the generalizability of genetic disease associations across populations, and the value that low-LD African ancestry individuals contribute to multi-ethnic fine-mapping of genetic associations. Despite the considerable federal support for the CAAPA initiative, we do recognize that the CAAPA sample size falls considerably short of the recent mega studies comprised of asthma datasets in the hundreds of thousands (5, 6). Furthermore, replication had to be sought in studies not primarily ascertained for asthma and with limited sample sizes. Nevertheless, we maintain that a failure to report on novel discoveries in this, the largest non-European asthma cohort to date, would result in social injustice. Important insights drawn from this study include the demonstration that many of the genetic loci associated with asthma in European ancestry populations may also be at play in African ancestry populations, and a clearer understanding of the LD patterns among African ancestry populations in 17q21. Potentially novel loci discovered by this meta-analysis are as yet not replicated within this study, but warrant follow-up. Importantly, the advent of institutional biobanks with access to multi-ethnic patient populations(7), as well as efforts by institutions such as the National Institute of Health to reduce health and research disparities(8) promise to greatly expand representation of well-characterized patients of African ancestry in the near future, allowing for robust follow-up of these CAAPA findings.

2) The rationale for using multiple comparisons correction statistical thresholds is reasonable as presented, but the need to employ different thresholds for different approaches is consistent with there not being strong and reliable association signals. For example, the EVE PYHIN1 association that was identified with some of the same cohorts that are part of CAAPA does not replicate in CAAPA. This suggests that a future CAAPA study with a larger sample size will find different associations than those listed here.

We thank the reviewer for noting that the statistical thresholds we applied are reasonable, as it is indeed standard practice to use a lower threshold when assessing loci identified by other GWAS. We used a stringent correction for claiming strong replication of the 810 TAGC SNPs that were formally tested for replication in CAAPA, i.e. we applied a Bonferroni correction for 810 tests, although the effective number of tests are likely <810 due to the correlation structure between SNPs located in the same locus. Although the chromosome 12q13 locus association could not be formally replicated in this manner, the lead SNP rs3122929 in CAAPA is in strong LD with the TAGC lead SNP rs167769 in Europeans from the TGP ($r^2=0.93$) and nearly achieves genome-wide significance in CAAPA ($p=9.1e-7$), which is strong evidence for replication. While we acknowledge that results novel to CAAPA are speculative until they can be replicated and/or validated, we believe that these are important results to record in the literature. The original EVE discovery GWAS was comparable in size to our GWAS (5,388 cases compared to our 7,009), and 3 loci (nearest genes IL1RL1, TSLP, GSDMB) crossed genome-wide significance ($P<5e-8$, (see table 2 of (9)). All three these loci were strongly associated with asthma in the recent TAGC paper, with $P<1e-19$ (see Table 1 of (5)). We therefore argue that it is plausible that our novel results, particularly the chr8p23 locus which passed genome-wide significance, will be replicated by future studies.

3) The 17q21 region results are interesting, and while asthma associations at this locus are not novel, understanding LD patterns among populations of different ancestry may help link specific variants to asthma. In addition to focusing on the SNPs listed in the Ober summary of this region, a closer comparison of the genome-wide significant SNPs in this region in CAAPA vs. those reported previously by TAGC would be of interest. Can TAGC results be presented along with those of Fig 2A? Figures that relate association results to the haplotype blocks among groups with more/less African ancestry as shown in Fig 2D and Fig S18.

We thank the reviewer for highlighting the need for this important comparison. We have updated Figure 2A to depict the 39 SNPs significant in TAGC Europeans that replicated in CAAPA (from Table S13) by squares. As the reviewer suggested, we have added the haplotype block of the population with the lowest and highest African ancestry to the bottom of Figure 2A, for these 39 SNPs. As the intent of Figure 2D was to demonstrate the breakdown of LD by degree of African ancestry in a main figure, and this information is now conveyed in Figure 2A (and the LD structure of the previous 22 SNPs for JAAS and GALA II is depicted in Figure S18), we have removed Figure 2D.

4) Until there are larger numbers of subjects of African Ancestry, comparing measures of association across racial/ethnic groups remains premature, but nonetheless, the finding that many of the European ancestry GWAS results are also observed in African Ancestry cohorts suggests that common genetic variation may not strongly contribute to asthma disparities. A discussion about the main message conveyed by title is lacking in the Discussion section.

We now discuss this issue in the first paragraph of the discussion.

It has been posited that disparities in asthma susceptibility can partly be explained by genetic risk factors (10-12). In recapitulating associations mainly discovered in European ancestry populations (a novel result in itself that has previously not been well-quantified in the literature), our results suggest that at the very least, common genetic variation may not strongly contribute to asthma disparities. However, our data shows the chromosome 17q12-21 associations have smaller effect sizes on African ancestry haplotypes. In addition, we built a genetic risk score for the 18 asthma loci reported by TAGC, and found that although asthmatics had a statistically significant higher risk score compared to controls (Supplementary Figure S24), the effect was too small to build a predictive risk score for asthma in CAAPA.

Minor

1) Suggested format change of P-values listed in Table 1 to not using scientific notation for $p > 10^{-3}$ We have reformatted the table accordingly.

Reviewer #2 (Remarks to the Author):

This paper describes the largest asthma GWAS to date in African ancestry populations. The paper is generally well written and conclusions are appropriate for the data presented. The following issues require attention.

1 Although this is the largest GWAS in African ancestry populations looking at asthma to date, the study size still remains modest, which probably explains the relative lack of either known or novel genome wide significant signals identified. Some known signals were identified including the well described chr17 locus, although only 2 reached conventional genome wide significance (17q and 8p23, although attempts to replicate the latter in smaller African ancestry populations were not successful). The results are hence rather incremental although still worthwhile to report.

We thank the reviewer for this thoughtful response to the challenging issue of sufficiently powered datasets of underrepresented minority populations. We have expanded the last paragraph of the discussion to further highlight the need to publish GWAS conducted in minority populations, and the important contribution of our study to the literature given the lack of data on genetic risk factors for asthma in African ancestry populations currently available. We wish to also underscore the additional penalty we face: in a climate where there is a dearth of federally funded research on underrepresented minorities, we were compelled to combine all available samples from African ancestry asthmatics and non-asthmatics into the discovery dataset, leaving limited samples available for replication. In short, we agree with the reviewer that our results are incremental, and that it is critical to report them in order to advance the field.

The first few sentences of the last paragraph of the Discussion now read as follows:

The CAAPA meta-analysis includes data from 15 independent studies and is the largest asthma GWAS focused on African ancestry populations to date. Unfortunately, as is the case for other complex diseases for which morbidity disproportionately affects underrepresented and underserved populations(1-3), a legacy of underrepresentation or exclusion of minorities from federally-funded studies has rendered comparatively robust non-European datasets rare to nonexistent. A recent analysis of ancestry represented in the GWAS Catalog(4) concluded that non-European, non-Asian groups combined account for less than 4% of individuals included in the catalog(1). The analysis also found that African ancestry individuals contributed 7% of all catalog associations, despite only comprising 2.4% of the catalog, highlighting the value of GWAS conducted in African ancestry populations for enabling scientific discoveries. Furthermore, the authors stressed the importance of assessing the generalizability of genetic disease associations across populations, and the value that low-LD African ancestry individuals contribute to multi-ethnic fine-mapping of genetic associations. Despite the considerable federal support for the CAAPA initiative, we do recognize that the CAAPA sample size falls considerably short of the recent mega studies comprised of asthma datasets in the hundreds of thousands (5, 6). Furthermore, replication had to be sought in studies not primarily ascertained for asthma and with limited sample sizes. Nevertheless, we maintain that a failure to report on novel discoveries in this, the largest non-European asthma cohort to date, would result in social injustice. Important insights drawn from this study include the demonstration that many of the genetic loci associated with asthma in European ancestry populations may also be at play in African ancestry populations, and a clearer understanding of the LD patterns among African ancestry populations in

17q21. Potentially novel loci discovered by this meta-analysis are as yet not replicated within this study, but warrant follow-up. Importantly, the advent of institutional biobanks with access to multi-ethnic patient populations(7), as well as efforts by institutions such as the National Institute of Health to reduce health and research disparities(8) promise to greatly expand representation of well-characterized patients of African ancestry in the near future, allowing for robust follow-up of these CAAPA findings.

2 I note the degree of African ancestry varied quite widely between the populations included. To deal with this the results were adjusted for the spectrum of ancestry in the meta-analysis. I was not entirely clear how this was done but reassurance that this approach did not potentially weaken signals seen in the populations with the highest extent of African ancestry would be useful.

A major goal of the CAAPA initiative was to capture the considerable genetic heterogeneity characteristic of admixed populations in general, and African admixed populations in particular. As noted in the first paragraph of the Results - Association Analysis section, we used the meta-analysis approach implemented in MR-MEGA, which “models allelic effects as a function of axes of genetic variation”, thus “heterogeneous associations across genetically distinct populations are not penalized”. MR-MEGA also models heterogeneity between studies unrelated to ancestry, analogous to random effects meta-analysis, which is appropriate for CAAPA due to differences in study design between studies. In order to demonstrate that signals in populations with the highest extent of African ancestry has not been weakened, we have added columns to Tables S4-S8 (reporting the top associations for populations representing low, intermediate and high African ancestry), and now report the overall (across all CAAPA studies) MR-MEGA association p-value, together with an overall traditional inverse-variance p-value. From this it can be seen that the top population-specific associations are of the same order in the CAAPA MR-MEGA and inverse-variance meta-analysis. Specifically, none of the results with population-specific p-values < 10⁻⁶ in intermediate or high African ancestry population have p-values < 10⁻⁶ in the CAAPA MR-MEGA or inverse-variance results, i.e. our choice of using MR-MEGA as a meta-analysis approach did not result in these associations not being detected in the meta-analysis.

It also appears that different chips were used for different populations and the imputed genotype results then meta-analysed: the imputation data used for this came from WGS data in relevant populations although again the degree of African ancestry varies somewhat in these populations.

Similar to previously reported meta-analyses leveraging existing GWAS data (e.g. 13 arrays for TAGC – Supplementary Table 2 of (5), 20 arrays for GIANT – Supplementary Table 18 of (13)), cohorts within CAAPA had previously been genotyped on a variety of commercial arrays. The CAAPA WGS imputation panel indeed varies in degree of African ancestry, and we have added estimates of African ancestry to Table S3, which summarizes the CAAPA WGS imputation panel. Also, the studies included in our meta-analysis did rely on different genotyping arrays (Table 1). Despite this heterogeneity, data was imputed with high accuracy, as described in Supplementary Note 8. We have also created a new supplementary table, Table S24, which summarizes the number of imputed SNPs that pass imputation quality filters. According to this table, at least 92% of common variants pass imputation quality filters in each study.

3 Whilst some data on the subjects included for study have been previously reported it would probably help readers of the manuscript if a brief summary of inclusion criteria for cases and controls was included in the supplementary methods (how was asthma defined?; are the controls age matched?).

This information is available in Supplementary Note 1, and we have revised the note with some additional information where inclusion criteria or asthma case-control definition was previously unclear. Controls were not matched by age (as indicated by the age distribution in Supplementary Table S1), and as such are not described in the note.

4 I assume additional phenotype data are also available eg on atopy and eosinophil counts: as some of the genetic associations between these phenotypes (eg eosinophil count and the IL1RL1 locus) overlap with asthma have these also been examined?

Unfortunately, eosinophil counts were only available on a small subset (~300 subjects from 2 studies). However, measures of total serum IgE (tgE), a quantitative trait that correlates with atopy and eosinophil counts, were available in ~4,000 subjects (this information is now included Supplementary Table S1). We tested SNPs reported for CAAPA and TAGC in Table 2 (see new heading “Tests for association with total serum IgE” in the online methods), and now report these results in the new Supplementary Table S20. Four of the SNPs tested are associated with tgE in asthmatics, having a direction of effect that is consistent with risk of asthma (higher tgE = increased risk of asthma, lower tgE = decreased risk of asthma), including the IL1RL1 locus. We have added the paragraph below to the results section, noting these results:

We also examined whether genetic associations with asthma overlap with atopy by testing lead SNPs from Table 2 for association with total serum IgE (tgE) using 4,132 subjects for whom this phenotype was available (CAAPA lead SNPs and lead SNPs from TAGC that replicated in CAAPA; associations were tested separately in cases and controls and then combined using meta-analysis). Four of the SNPs correlated with levels of tgE in asthmatics (with the asthma risk allele associated with increased levels of tgE, Supplementary Table S20), suggesting that these asthma risk alleles may lead to an increased Th2 immune response. This includes SNP rs1420101 in the IL1RL1 locus (a gene that is known to correlate with eosinophilia (14)), SNP rs10519067 intronic to the RORA gene (encoding a transcription factor that regulates the growth of group 2 innate lymphoids, a key cell type in the memory Th2 cell response (15)), and SNPs rs907092 and rs2952156 in the chr17q12-21 locus. These results are consistent with previous studies that have shown both shared and unique associations between the phenotypes (16-20).

5 I note that an attempt to look at known associations has been made mostly on a SNP by SNP basis. What would be most interesting is to try and look at whether the genetic landscape of asthma is similar in African ancestry individuals or actually quite different. Whilst the study lacks much power to look at this, it should be possible to use TAGC data to generate an asthma risk score using multiple

SNPs having genome wide significant effects, and then to look at how this risk score predicts asthma in the African ancestry population. One minor issue here is that some of the individuals used in the current analysis were included in TAGC analyses although they form a small proportion of the 142k individuals in TAGC.

We intersected the genome-wide significant SNPs in TAGC Europeans (thereby not including subjects also included in CAAPA) with SNPs available in all the CAAPA studies and selected the lead (most significant) SNP in each of the broader 18 loci to build a risk score in the CAAPA studies. These results are presented in Supplementary Figure S24, and we briefly refer to these results in the first paragraph of the Discussion. Overall, asthmatics had a statistically significant higher risk score compared to controls, although the effect was not statistically significant in many of the individual studies, and the effect was too small to build a predictive risk score. Ideally, to test whether the landscape of asthma is different in African ancestry populations, we plan to eventually develop a risk score built from the CAAPA meta-analysis results, to be tested in the CAAPA replication samples, alongside the TAGC genetic risk score for comparative purposes. This is unfortunately not possible at this time as our investigative team does not have access to the full genetic data sets of the replication studies.

6 It is suggested that the potentially novel (but not replicated) 8p23 signal could be driven by variants in either ARHGEF10 or MYOM2: a little more detail on how the credible set of variants was identified for eQTL look ups and what expression data were then used to define a credible set of genes would be helpful.

We thank the reviewer for highlighting that these crucial details are lacking. To address this, we have added Supplementary Note 13, describing how the credible set of SNPs were defined, and list these SNPs in the new Supplementary Table S23. The note describes our attempts to integrate information from eQTL databases as well as ENCODE. These attempts were unfruitful, and we therefore used the HUG-In tool to visualize long-range chromatin interactions, as the observed associations could be due to the regulation of genes not in the immediate vicinity of the association signal. We have also described the tool in more detail in the Supplementary Note 13, as well as our interpretation of its visualization. We refer the reader to this more detailed description of the analysis in the main text (added wording in bold font):

While the associated SNPs in this region do not overlap with any expression quantitative trait locis (eQTLs) in the publicly available databases we mined, long-range chromatin interaction and expression data in relevant tissues (lymphoblastoid cells, fetal lung fibroblast cells, lung) implicate two different genes, ARHGEF10 and MYOM2, ~600-800 KB downstream from the most significant SNP rs13277810, that potentially explain these observed associations (Supplementary Note 13, Supplementary Figure S19)

Reviewer #3 (Remarks to the Author):

The paper by Daya et al describes the largest genome-wide meta-analysis for asthma in African ancestry individuals. Undertaking such studies in non-European ancestry groups is going to be crucial in bringing genetically-driven medical advances to these populations, and the authors should be commended for the scale and ambition of the study as well as for the rigour with which the study was undertaken. Whilst the study yields little in the way of novel signals meeting the most stringent genome-wide levels of significance and with convincing replication, it nevertheless yields important insights and it undoubtedly represents a major step from studies of this kind previously undertaken. The admixture mapping adds a novel element. It will be of interest to the scientific community, especially those with particular interests in genetics understanding of disease in non-European ancestry populations or in respiratory health and disease.

Major comments

1. Replication:

a. Of the two new loci on Chromosome 8, the evidence from replication studies is presented in Suppl Table 14. This should be moved to the main paper so that the reader can evaluate the evidence for themselves.

We thank the reviewer for this suggestion, and agree that this is important information for the reader to evaluate. We have moved Table S14 to the main text as Table 3.

b. Results from BioMe do support the first of these signals (but were unavailable for the second). One feature not discussed is that CARDIA, MESA and ARIC differ quite fundamentally from the discovery studies in having a much larger number of asthma cases than controls (18 times as many cases for ARIC), whereas the discovery cohorts had more controls than cases. Might the results in the replication cohorts be subject to ascertainment bias?

We very much agree with this point, and have revised the second paragraph of the discussion to highlight the relatively small number of asthmatics available for replication, and differences in ascertainment. The second paragraph of the Discussion now reads as follows:

In addition to recapitulating asthma genes discovered largely in non-African populations, we identified two novel loci on chromosome 8, and through admixture mapping identified a novel region on chromosome 6q22. The most significant SNP on chromosome 8p23 reached genome-wide significance and was replicated in Puerto Ricans from BioMe. However, our attempts to replicate this same locus in BioMe African Americans were unsuccessful, likely due to small sample size, and the relatively small number of cases compared to controls (4 to 18 times smaller), as the replication studies were not ascertained for asthma. Similarly, we also failed to replicate the low frequency variant on chromosome 8q24 (only 398 asthmatics, all African American, were available to replicate this low frequency variant) and the admixture mapping signal (only 845 cases were available, of which 498 were African American). While we cannot claim

replication, possibly due to small samples sizes and differences in ascertainment, these findings may still be true associations, and warrant further replication efforts.

c. The results from these studies do not support replication but these analyses would be underpowered compared to BioMe, for example. My interpretation of the analyses would be that the 8p23 signal is supported and that adequate replication for the 8p24 signal is still required. This could be covered more clearly in the discussion.

As stated above in our response to 1b, we have revised the second paragraph of the discussion, highlighting that the 8p23 signal replicated in BioMe, and that the 8p24 replication sample size is inadequate.

d. I am not concerned about the results being reported without further replication – the limitation here is the availability of relevant datasets and the authors should be commended for the analysis.

We thank the reviewer for this thoughtful response and importantly, recognition of the inherent challenges owed to a legacy of underrepresentation of African ancestry individuals in genetic studies of complex traits. To underscore this point, we have expanded the last paragraph of the discussion to further highlight that our study makes an important contribution to the literature due to the lack of data on genetic risk factors for asthma in African ancestry populations currently available.

2. The reader would benefit from greater assistance to navigate through the large amount of information in the main text and supplement and improved connectivity between elements. For example, where cohorts are referred to in the main text by geographical origin e.g. lines 322-323, or by cohort name only (Figure 1A) it would be easier if the two were put together. Perhaps the information on location from Table S1 could be moved to the main text and figure 1A annotated by geographical origin. It would make it much easier to see at a glance which cohorts included African American samples.

We thank the reviewer for their helpful comments and suggestions for improving navigation through the manuscript. We have updated Figure 1A to also include geographical information. Table S1 is now included as Table 1 (and the original Table 1 has been renamed to Table 2). We have also updated the main manuscript to always state the geographical location of studies in brackets.

3. Whilst it is reasonable that some of the cohorts have been included in a previous meta-analysis (TAGC), it would be helpful for the reader to see (ideally in the main text) what the overlap between TAGC and the current (CAAPA) analysis. The authors do appear to have removed overlapping studies when making key comparisons.

Because some of the CAAPA studies were included in TAGC, all the key comparisons were made using the European only TAGC association results so that we could avoid overlap in subjects. In the main text, under the section headed “Comparison to previous asthma GWAS”, we have now added the following information on the overlap between TAGC and CAAPA:

of the 2,149 asthmatics and 6,055 non-asthmatics in TAGC, 1,601 asthmatics (74%) and 2,375 non-asthmatics (39%) were from studies included in the CAAPA discovery, and 548 asthmatics and 3,680 non-asthmatics were from studies included in the CAAPA replication

Minor comments

Line 43 – show (rather than “shows”) *Corrected*

Novel line 104 – novel in which study “a novel locus on chromosome 12q13 not yet replicated” has been rephrased as “a locus on chromosome 12q13 reported as novel by TAGC that has not yet been replicated”

Line 105 – missing “8” after chromosome? *8 was added after chromosome*

Line 128 – omit “critical” (I don’t see that this threshold is critical) *We have omitted critical*

Line 139 –missing “gene encoding”? *We have added gene encoding*

Line 209-221: Is the Chromosome 12 signal the same as one previously described for lung function (FEV1/FVC, Soler Artigas et al Nat Genet 2011) at LRP1? STAT6 (mentioned on line 213) was described as one of the “plausible genes” for this locus in Wain et al Nat Genet 2017 (suppl Table 15).

We thank the reviewer for highlighting this previous report, and yes indeed, it is the same locus! We have revised these lines to now read as follows (added text in bold font):

*The TAGC lead SNP rs167769 on chromosome 12q13 is intronic to STAT6, a transcription factor that affects Th2 lymphocyte responses mediated by IL-4 and IL-13 (5, 21). This was a new association reported by TAGC, not previously implicated in any asthma GWAS, although we note **SNP rs167769 has been reported as a putatively causal SNP discovered by GWAS of lung function (22, 23), and a number of linkage studies have pinpointed this chromosomal region in the early days of genome-wide investigations of asthma and atopy (24-27).***

Perhaps the authors can comment in the methods or discussion on the power for admixture mapping in discovery and replication.

We have added a section in the online methods on admixture mapping power. While we are adequately powered to detect relative risks ≥ 1.2 in both the discovery and replication, given the effect size of the lead segment in the discovery (local ancestry ratio of 1.12), we are underpowered to replicate the signal. We now note this in the admixture mapping results section of the main manuscript (closing sentence of the paragraph under the "Replication" heading):

However, we note that our replication sample size is likely underpowered (see Online Methods).

1. Morales J, Welter D, Bowler EH, Cerezo M, Harris LW, McMahon AC, et al. A standardized framework for representation of ancestry data in genomics studies, with application to the NHGRI-EBI GWAS Catalog. *Genome biology*. 2018;19(1):21.
2. West KM, Blacksher E, Burke W. Genomics, Health Disparities, and Missed Opportunities for the Nation's Research Agenda. *JAMA*. 2017;317(18):1831-2.
3. Popejoy AB, Fullerton SM. Genomics is failing on diversity. *Nature*. 2016;538(7624):161-4.
4. MacArthur J, Bowler E, Cerezo M, Gil L, Hall P, Hastings E, et al. The new NHGRI-EBI Catalog of published genome-wide association studies (GWAS Catalog). *Nucleic Acids Res*. 2017;45(D1):D896-D901.
5. Demenais F, Margaritte-Jeannin P, Barnes KC, Cookson WOC, Altmüller J, Ang W, et al. Multiancestry association study identifies new asthma risk loci that colocalize with immune-cell enhancer marks. *Nature Genetics*. 2018;50(1):42-53.
6. Ferreira MA, Vonk JM, Baurecht H, Marenholz I, Tian C, Hoffman JD, et al. Shared genetic origin of asthma, hay fever and eczema elucidates allergic disease biology. *Nat Genet*. 2017;49(12):1752-7.
7. Wolford BN, Willer CJ, Surakka I. Electronic health records: the next wave of complex disease genetics. *Hum Mol Genet*. 2018.
8. Health Disparities. National Institute of Health; 2010.
9. Torgerson DG, Ampleford EJ, Chiu GY, Gauderman WJ, Gignoux CR, Graves PE, et al. Meta-analysis of genome-wide association studies of asthma in ethnically diverse North American populations. *Nat Genet*. 2011;43(9):887-92.
10. Gould W, Peterson EL, Karungi G, Zoratti A, Gaggin J, Toma G, et al. Factors predicting inhaled corticosteroid responsiveness in African American patients with asthma. *J Allergy Clin Immunol*. 2010;126(6):1131-8.
11. Mathias RA, Grant AV, Rafaels N, Hand T, Gao L, Vergara C, et al. A genome-wide association study on African-ancestry populations for asthma. *J Allergy Clin Immunol*. 2010;125(2):336-46 e4.
12. Sulovari A, Chen YH, Hudziak JJ, Li D. Atlas of human diseases influenced by genetic variants with extreme allele frequency differences. *Hum Genet*. 2017;136(1):39-54.
13. Wood AR, Esko T, Yang J, Vedantam S, Pers TH, Gustafsson S, et al. Defining the role of common variation in the genomic and biological architecture of adult human height. *Nat Genet*. 2014;46(11):1173-86.
14. Esnault S, Kelly EA, Schwantes EA, Liu LY, DeLain LP, Hauer JA, et al. Identification of genes expressed by human airway eosinophils after an in vivo allergen challenge. *PLoS One*. 2013;8(7):e67560.

15. Lima LC, Queiroz GdA, Costa RdS, Alcantara-Neves NM, Marques CR, Costa GNdO, et al. Genetic variants in RORA are associated with asthma and allergy markers in an admixed population. *Cytokine*. 2018.
16. Sunyer J, Anto JM, Castellsague J, Soriano JB, Roca J. Total serum IgE is associated with asthma independently of specific IgE levels. The Spanish Group of the European Study of Asthma. *Eur Respir J*. 1996;9(9):1880-4.
17. Sears MR, Burrows B, Flannery EM, Herbison GP, Hewitt CJ, Holdaway MD. Relation between airway responsiveness and serum IgE in children with asthma and in apparently normal children. *N Engl J Med*. 1991;325(15):1067-71.
18. Palmer LJ, Burton PR, Faux JA, James AL, Musk AW, Cookson WO. Independent inheritance of serum immunoglobulin E concentrations and airway responsiveness. *Am J Respir Crit Care Med*. 2000;161(6):1836-43.
19. Levin AM, Mathias RA, Huang L, Roth LA, Daley D, Myers RA, et al. A meta-analysis of genome-wide association studies for serum total IgE in diverse study populations. *J Allergy Clin Immunol*. 2013;131(4):1176-84.
20. Burrows B, Martinez FD, Halonen M, Barbee RA, Cline MG. Association of asthma with serum IgE levels and skin-test reactivity to allergens. *N Engl J Med*. 1989;320(5):271-7.
21. Goenka S, Kaplan MH. Transcriptional regulation by STAT6. *Immunol Res*. 2011;50(1):87-96.
22. Wain LV, Shrine N, Artigas MS, Erzurumluoglu AM, Noyvert B, Bossini-Castillo L, et al. Genome-wide association analyses for lung function and chronic obstructive pulmonary disease identify new loci and potential druggable targets. *Nat Genet*. 2017;49(3):416-25.
23. Soler Artigas M, Loth DW, Wain LV, Gharib SA, Obeidat M, Tang W, et al. Genome-wide association and large-scale follow up identifies 16 new loci influencing lung function. *Nat Genet*. 2011;43(11):1082-90.
24. Barnes KC, Neely JD, Duffy DL, Freidhoff LR, Breazeale DR, Schou C, et al. Linkage of asthma and total serum IgE concentration to markers on chromosome 12q: evidence from Afro-Caribbean and Caucasian populations. *Genomics*. 1996;37(1):41-50.
25. A genome-wide search for asthma susceptibility loci in ethnically diverse populations. The Collaborative Study on the Genetics of Asthma (CSGA). *Nat Genet*. 1997;15(4):389-92.
26. Nickel R, Wahn U, Hizawa N, Maestri N, Duffy DL, Barnes KC, et al. Evidence for linkage of chromosome 12q15-q24.1 markers to high total serum IgE concentrations in children of the German Multicenter Allergy Study. *Genomics*. 1997;46(1):159-62.
27. Barnes KC, Freidhoff LR, Nickel R, Chiu YF, Juo SH, Hizawa N, et al. Dense mapping of chromosome 12q13.12-q23.3 and linkage to asthma and atopy. *J Allergy Clin Immunol*. 1999;104(2 Pt 1):485-91.

Reviewer #1 (Remarks to the Author):

Summary

This resubmission by Daya et al is improved with the addition of further discussion and analyses. The initial assessment that results presented are interesting but largely descriptive still applies, but I agree with authors that publication of results is worthwhile. New issues to address follow.

Major

- 1) The sentence “Nevertheless, we maintain that a failure to report on novel discoveries in this, the largest non-European asthma cohort to date, would result in social injustice.” should be struck from the discussion section. While the passion of authors is admirable, the social injustice lays in the exclusion of groups from studies in the first place, not in the publication of this specific manuscript.
- 2) The LD plots in Figure 2 are interesting although not much can be concluded about specific SNP association differences by race/ethnicity, other than to say that LD in this region is considerably different. The statements in lines 260-268 that effect sizes are proportional to YRI ancestry and/or may be due to Native American ancestry are not supported by Figure 2B. I fail to see any trend in the forest plot, and if Native American ancestry is causing an exception for HONDAS and GALA2, then how would authors explain PGCA? This section should be corrected. What the LD plot shows clearly is that a region of high LD in populations of European ancestry would result in similar associations for the whole region, while the heterogeneous LD in populations of African ancestry would have different association signals in this region, depending on which SNP is studied.
- 3) Similarly, Figure 2C is not very compelling in supporting the statement that African ancestry decreases effect size. Perhaps there is a trend between 0 copies vs. 1 and 2, but there is simply too much variability in the box plots, with marginal overall changes in OR for this effect to be appreciable (mean ~ 1.3 vs. 1.2). A statistical test could be provided to support the claim that the ORs are different in each group, but sentences should be rephrased to state that a trend was noted and an unproven hypothesis was generated on this basis.

Minor

- 1) Statements should be made for why individuals are in the acknowledgments section

Reviewer #2 (Remarks to the Author):

The authors have provided a thoughtful response to the issues raised and made changes to improve the manuscript.

Reviewer #3 (Remarks to the Author):

I am satisfied that the reviewers' comments have been addressed.

Our response to the comments from the reviewers are in blue font, added below each relevant comment.

Reviewer #1 (Remarks to the Author):

Summary

This resubmission by Daya et al is improved with the addition of further discussion and analyses. The initial assessment that results presented are interesting but largely descriptive still applies, but I agree with authors that publication of results is worthwhile. New issues to address follow.

Major

1) The sentence “Nevertheless, we maintain that a failure to report on novel discoveries in this, the largest non-European asthma cohort to date, would result in social injustice.” should be struck from the discussion section. While the passion of authors is admirable, the social injustice lays in the exclusion of groups from studies in the first place, not in the publication of this specific manuscript.

We have removed this sentence from the discussion.

2) The LD plots in Figure 2 are interesting although not much can be concluded about specific SNP association differences by race/ethnicity, other than to say that LD in this region is considerably different. The statements in lines 260-268 that effect sizes are proportional to YRI ancestry and/or may be due to Native American ancestry are not supported by Figure 2B. I fail to see any trend in the forest plot, and if Native American ancestry is causing an exception for HONDAS and GALA2, then how would authors explain PGCA? This section should be corrected. What the LD plot shows clearly is that a region of high LD in populations of European ancestry would result in similar associations for the whole region, while the heterogeneous LD in populations of African ancestry would have different association signals in this region, depending on which SNP is studied.

We agree with the reviewer that differing patterns in LD as well as higher European ancestry may be in part responsible for the patterns we are observing. However, we speculate that this is most likely due to the Native American component as we observed this strong pattern in HONDAS where there is very minimal European ancestry. We have edited the section on this as detailed below, and hope this suitably addresses the concerns:

Associations between SNPs on chr17q12-21 and risk of asthma showed evidence of ancestry heterogeneity (46 of the 54 SNPs with association $p < 10^{-6}$ had significant heterogeneity, $p < 0.05$). In general, the magnitude of the effect size increases as the average proportion of European ancestry in the study increased (Supplementary Table 9; most of the corresponding β_0 values were close to zero, whereas β_1 , which captures the effect along the axis of genetic variation separating African and European ancestry, showed an increase in effect size magnitude as European ancestry increased). We also observe that the higher average proportion of African ancestry, the smaller the magnitude of the effect size (Figure 2B, forest plot of the lead ordered by average African ancestry). An exception was observed among Honduran subjects who self-reported as Garifuna (HONDAS). HONDAS has a large average African (77%), a higher Native American (20%), and a very small European component (3%). Interestingly, the lead SNP in CAAPA was the same SNP reported by a meta-analysis of asthma in Puerto Rican children [Yan et al. 2017], distinct from the lead SNPs reported by the multi-ethnic EVE and TAGC GWAS [Torqerson et al 2011, Demenais et al. 2018]. The most significant and largest effect size magnitudes were observed for the studies that had higher European and Native American components, however we speculate that this may reflect risk for asthma inherited from a Native American genetic background given the minimal European component in the HONDAS population. We note this trend is not as strong in PGCA, the CAAPA study with the highest proportion of Native American ancestry (29%), which may be due in part to the heterogeneous patterns of LD in this region.

3) Similarly, Figure 2C is not very compelling in supporting the statement that African ancestry decreases effect size. Perhaps there is a trend between 0 copies vs. 1 and 2, but there is simply too much variability in the box plots, with marginal overall changes in OR for this effect to be appreciable (mean ~1.3 vs. 1.2). A statistical test could be provided to support the claim that the ORs are different in each group, but sentences should be rephrased to state that a trend was noted and an unproven hypothesis was generated on this basis.

Due to the correlation structure between SNPs in this region, which is further complicated by differences in the correlation structure based on ancestry and a small sample size of only 22 SNPs, we did not perform a statistical test to measure differences in effect size. Per the reviewers' suggestion, we have modified the last sentences of the relevant paragraph to now read as follows (changed/added text are underlined):

Figure 2C shows a trend of decreasing effect size as the number of copies of African ancestry increased, for most of these 22 candidate SNPs, suggesting (but not conclusively proving) smaller effect sizes on African ancestry haplotypes.

Minor

1) Statements should be made for why individuals are in the acknowledgments section

We have added this information, and the acknowledgement section now reads as follows:

We thank Goncalo Abecasis for coordinating inclusion of the CAAPA reference panel on the Michigan Imputation Server, Todd Deppe, Estelle Giraud, Cindy Lawley from Illumina for genotyping services, and Pat Oldewurtel for administrative support.

Reviewer #2 (Remarks to the Author):

The authors have provided a thoughtful response to the issues raised and made changes to improve the manuscript.

Reviewer #3 (Remarks to the Author):

I am satisfied that the reviewers' comments have been addressed.